The Venice specimen of Ouranosaurus nigeriensis (Dinosauria, Ornithopoda)

Bertozzo Filippo filippo.bertozzo@gmail.com 1
Dalla Vecchia Fabio Marco 2 4
Fabbri Matteo 3
1 Earth System Science-AMGC, Vrije Universiteit Brussel , Pleinlaan 2, Brussels/Bruxelles , Belgium
2 Institut Català de Paleontologia Miquel Crusafont, Sabadell , Spain
3 Department of Geology and Geophysics, Yale University , New Haven , CT , United States of America
4 Current affiliation:  Soprintendenza Archeologia, Belle Arti e Paesaggio del Friuli Venezia Giulia, Nucleo Operativo di Udine , Udine , Italy
Sues Hans-Dieter
Electronic publication date: 2017 Jun 20
Publication date: 2017
Volume: 5
Electronic Location ID: e3403
Received 2016 Nov 16; Accepted 2017 May 10
Copyright: ©2017 Bertozzo et al.
Copyright year: 2017
Copyright holder: Bertozzo et al.
License: This is an open access article distributed under the terms of the Creative Commons Attribution License, which permits unrestricted use, distribution, reproduction and adaptation in any medium and for any purpose provided that it is properly attributed. For attribution, the original author(s), title, publication source (PeerJ) and either DOI or URL of the article must be cited.
License URL: https://creativecommons.org/licenses/by/4.0/

Keywords: Ouranosaurus nigeriensis, Styracosterna, Ornithopoda, Dinosauria, Gadoufaoua, Gondwana, Cretaceous

Funding: Museo di Storia Naturale di Venezia (MSNVE) This study was partially supported by the Museo di Storia Naturale di Venezia (MSNVE). There was no additional external funding received for this study. The funders had no role in study design, data collection and analysis, decision to publish, or preparation of the manuscript.

==============================
Ouranosaurus nigeriensis is an iconic African dinosaur taxon that has been described on the basis of two nearly complete skeletons from the Lower Cretaceous Gadoufaoua locality of the Ténéré desert in Niger. The entire holotype and a few bones attributed to the paratype formed the basis of the original description by Taquet (1976). A mounted skeleton that appears to correspond to O. nigeriensis has been on public display since 1975, exhibited at the Natural History Museum of Venice. It was never explicitly reported whether the Venice specimen represents a paratype and therefore, the second nearly complete skeleton reported in literature or a third unreported skeleton. The purpose of this paper is to disentangle the complex history of the various skeletal remains that have been attributed to Ouranosaurus nigeriensis (aided by an unpublished field map of the paratype) and to describe in detail the osteology of the Venice skeleton. The latter includes the paratype material (found in 1970 and collected in 1972), with the exception of the left femur, the right coracoid and one manus ungual phalanx I, which were replaced with plaster copies, and (possibly) other manus phalanges. Some other elements (e.g., the first two chevrons, the right femur, the right tibia, two dorsal vertebrae and some pelvic bones) were likely added from other individual/s. The vertebral column of the paratype was articulated and provides a better reference for the vertebral count of this taxon than the holotype. Several anatomical differences are observed between the holotype and the Venice specimen. Most of them can be ascribed to intraspecific variability (individual or ontogenetic), but some are probably caused by mistakes in the preparation or assemblage of the skeletal elements in both specimens. The body length of the Venice skeleton is about 90% the linear size of the holotype. Osteohistological analysis (the first for this taxon) of some long bones, a rib and a dorsal neural spine reveals that the Venice specimen is a sub-adult; this conclusion is supported by somatic evidence of immaturity. The dorsal ‘sail’ formed by the elongated neural spines of the dorsal, sacral and proximal caudal vertebrae characterizes this taxon among ornithopods; a display role is considered to be the most probable function for this bizarre structure. Compared to the mid-1970s, new information from the Venice specimen and many iguanodontian taxa known today allowed for an improved diagnosis of O. nigeriensis.

Introduction

Ouranosaurus nigeriensis and Spinosaurus aegyptiacus are iconic African dinosaurs because of their common possession of hypertrophic neural spines. O. nigeriensis comes from the upper part of the El Rhaz Formation at the Gadoufaoua locality of the Sahara Desert, located 145 km east of Agadez, Niger (Taquet, 1976). The El Rhaz Formation of Niger has yielded a rich dinosaur fauna including theropods (Suchomimus, Cristatosaurus, Kryptops and Eocarcharia), sauropods (Nigersaurus) and the ornithopods Ouranosaurus and Lurdusaurus (LeLoeuff et al., 2012). It was considered to be Aptian by Taquet (1976) and Aptian–Albian by Sereno et al. (1999), but LeLoeuff et al. (2012) have proposed a Barremian age.

The only detailed anatomical description of O. nigeriensis was published by Taquet (1976), with only a few specimens formally referred to the hypodigm of O. nigeriensis: the holotype skeleton GDF 300, the paratype skeleton GDF 381 and two isolated bones (GDF 301 and 302; Taquet, 1976, p. 58), although the discovery of several additional in situ skeletons is mentioned in that paper (Taquet, 1976, p. 14–15). No other scientific works have been dedicated to the description of this taxon since 1976, although a few papers referred to it (Rasmussen, 1998; Dean-Carpentier, 2008; Taquet, 2012). Despite the difficulty of gaining access to study the original holotype material, it is always included in cladistic analyses of iguanodontian dinosaurs (e.g., Sereno, 1986; Norman, 2004; Norman, 2015; McDonald, Barrett & Chapman, 2010; McDonald, Wolfe & Kirkl, 2010; McDonald et al., 2012b; Wang et al., 2013; Tsogtbaatar et al., 2014).

Since 1975, a nearly complete mounted skeleton of O. nigeriensis has been exhibited at the Museo di Storia Naturale (Natural History Museum) of Venice, Italy. Apparently, the Venice specimen is not referred to by Taquet (1976) or in any other scientific papers dealing with Ouranosaurus. Therefore, this skeleton can be considered as undescribed.

The aim of this research was to uncover the following: (1) to disentangle the history of the discovery of Ouranosaurus skeletons with a particular focus on the Venice specimen; (2) to describe the latter and compare its osteology with Taquet (1976); (3) to perform the first osteohistological analysis on O. nigeriensis by sampling a variety of bones to determine the ontogenetic stage of the Venice specimen; (4) to establish whether the bones of the nearly complete mounted skeleton can be reliably referred to a single individual or whether it is composed of several individuals.

Materials and Methods

The focus of this paper is the Venice specimen of O. nigeriensis (MSNVE 3714; Fig. 1). It appears to be a nearly complete skeleton mounted in a bipedal posture. The specimen was donated to the MSNVE by the Italian entrepreneur and philanthropist Giancarlo Ligabue (founder of the Centro Studi e Ricerche Ligabue, Venice), who passed away in 2015. According to the available information, the specimen underwent two distinct restoration phases. The first preparation of the bones used in the mount was done by French preparators at the Muséum National d’Histoire Naturelle of Paris before 1975 when the skeleton was mounted in Venice. The specimen was restored, casted and remounted by an Italian private firm in 1999–2000. No reports or any kind of documentation exist about the restoration of the bones. A list of the original material does not exist. Ronan Allain, MNHN, kindly made available to us a copy of the field map of the in situ specimen, which was drawn by Philip Taquet and is stored at MNHN.

Figure 1 MSNVE 3714, Ouranosaurus nigeriensis.

The mounted specimen as exhibited today at the MSNVE. For scale, the right femur is 920 mm long.

This specimen is mounted on a metal frame and in order to photograph and describe it, all of the bones were dismounted from the frame, with the exception of the sacrum, which is fixed to the frame. Pictures of every original bone in its cranial (anterior), caudal (posterior), dorsal, ventral, lateral, and medial views were taken, using a Canon EOS 600D camera with 100 ISO sensitivity and a Tamron 17–50 mm (F/2.8) lens at focal distance 50 mm. The photographs are stored in the archive of the MSNVE, which is accessible to researchers by contacting the responsible people for Research and Scientific Divulgation of the Museum. A 200 mm-long caliper with measurement error of 0.01 mm and a 100 cm-long metric string (measurement error of 0.1 cm) were used to measure the bones. A table with all the measurements is included in the Supplemental Information 1-2. In order to identify the reconstructed parts, we took pictures under UV-light using a Wood Lamp (SKU 51029, emitting ultraviolet light at 4 W).

Caudal vertebrae with pleurapophyses (often reported as caudal ribs or transverse processes in the literature) are considered as proximal caudals; those lacking pleurapophyses but with haemapophyses are middle caudals; distal caudals lack pleurapophyses and hemapophyses. The cervical-dorsal transition in the vertebral column was identified following Norman (1986). The height of the centrum was measured at the caudal (posterior) articular facet, and the height of the neural spine was measured as the straight line from the mid-point of the spine in correspondence with the dorsal margin of the postzygapophysis to the apex of the spine (see Supplemental Information 1).

Bone surface texture, degree of fusion of the elements and obliteration of the sutures in skulls and vertebrae are the most common approaches to assess the ontogenetic stage of fossil tetrapods (e.g., Bennett, 1993; Brochu, 1996; Werning, 2012). However, histological analysis remains the most reliable methodology for establishing osteological maturity or immaturity and for estimating the absolute age of an individual (e.g., Erickson et al., 2004; Chinsamy, 2005; Erickson, 2005). The left humerus, right femur, right tibia, neural spine of dorsal vertebra 14 and right dorsal rib 15 were selected for osteohistological analysis. Core samples were taken from the long bones following the method described by Stein & Sander (2009), using an electric drill press Timbertech Kebo01 and a cylindrical diamond drill bit (16 mm in diameter, 80 mm in height and with a 2 mm-thick wall). Samples were taken from the diaphysis of the long bones. Only areas lacking evident superficial erosion and surface cracks were selected. The proximal shaft of the rib was cut transversely. That area was selected because it is considered to preserve the most complete growth record (Erickson, 2005). The neural spine was cross-sectioned at three different levels: at the base, in the middle, and in the apical region. Samples were then mounted on glass slides, polished to a thickness of ∼70 µm and analyzed with Leica DMLP and Nikon Optiphot2-pol microscopes. The type of microstructure, the density and type of vascular canals, the amount of remodeling, the number of Lines of Arrested Growth (LAGs) and the presence or absence of an External Fundamental System (EFS) are the proxies used in this study to evaluate the ontogenetic stage of the sampled skeletal elements. The definition of the type of arrangement of the vascular canals was based on the orientation of their main axis. LAGs were identified and counted when an arrest in bone deposition was visible at different magnifications and when the interruption was continuous along the slide. When two or more LAGs were tightly spaced in the inner cortex, these were considered as annuli and counted as a single year.

Results

Historical background of the Venice specimen

According to Ligabue & Rossi-Osmida (1975) and Bonaparte et al. (1984, p. 310), the Venice specimen was collected by an Italian-French expedition lead by G Ligabue and P Taquet in 1973. However, no reference to that expedition and that O. nigeriensis specimen can be found in Taquet (1976), which mentions only two skeletons: the holotype (now at the MNBH) and the paratype. According to Currie & Padian (1997, p. 369), the only original specimen of O. nigeriensis other than the holotype is the Venice specimen, indirectly suggesting that it is the paratype. In order to establish whether the Venice specimen is the paratype or another skeleton, the complicated historical background of Ouranosaurus discoveries had to be disentangled based on the literature (Ligabue et al., 1972; Ligabue & Rossi-Osmida, 1975; Taquet, 1976; Taquet, 1998; Boccardi & Bottazzi, 1978; Bonaparte et al., 1984), the information available at the MSNVE and personal communications with P Taquet.

Between 1965 and 1972, five French palaeontological expeditions searched for dinosaurs in the Gadoufaoua area of the Sahara desert in Niger (Taquet, 1976). The first expedition took place in January–February 1965, resulting in the discovery of eight iguanodontian specimens at the site “niveau des Innocents”, located east of the Emechedoui wells. Two further iguanodontian skeletons, labelled GDF 300 and GDF 381, were found 7 km south-east of Elrhaz in the Camp des deux arbres locality.

During the second expedition (February 25th–April 7th, 1966), GDF 300 (a nearly complete but disarticulated and scattered skeleton) and GDF 381 (“a skeleton two thirds complete”, p. 54) were collected. The following year, those specimens were carried to Paris for preparation and study. GDF 300 became later the holotype of Ouranosaurus nigeriensis (Taquet, 1976, p. 57). The other specimen (GDF 381), which was found 100 m from GDF 300 and is referred to as the “Iguanodontidé trapu (ponderous Iguanodontid)” by Taquet (1976, p. 54; see also p. 14 and 53), subsequently became the holotype of Lurdusaurus arenatus (see Taquet & Russell, 1999; P Taquet, pers. comm., 2012). However, the holotype skeleton of L. arenatus received the new number MNHN GDF 1700 once in Paris, while the previous field number GDF 381 remained associated with an isolated right coracoid that was referred to the same species (Taquet & Russell, 1999, p. 3).

The third expedition (1969) found some dinosaur material at the In Gall locality (actually outside the Gadoufaoua area), but no Ouranosaurus skeleton is reported from there.

During the fourth expedition (January 5th–March 23rd, 1970), a nearly complete O. nigeriensis skeleton lacking its skull, but in better state of articulation than GDF 300, was discovered 4 km south of the “niveau des Innocents” at the margin of the landing strip built by the CEA, (p. 58). This skeleton also received the field number GDF 381 (see Taquet, 1976, pl. IX, fig. 2). So, the field number GDF 381 was erroneously used three times to indicate three different specimens found in different years. This nearly complete skeleton without skull was later indicated as the paratype of O. nigeriensis and reported by Taquet (1976) as GDF 381-MNHN on p. 58 and as GDF 381 throughout the text.

During the fifth expedition (January 5th–February 25th, 1972), the Ouranosaurus skeleton found in 1970 (i.e., the paratype) was excavated and brought to Paris (Taquet, 1976, p. 15 and 60). Apparently, this is the third and last ornithopod skeleton from Gadoufaoua excavated and brought to France by French expeditions, together with the holotypes of O. nigeriensis and L. arenatus found in 1965 and collected in 1966.

In 1971, Giancarlo Ligabue and Cino Boccazzi knew about the Gadoufaoua locality while travelling across the Sahara desert (Ligabue et al., 1972). Ligabue and the CNR financially supported the first Italian expedition (February 3rd–22nd, 1972; at the same time as the fifth French expedition), which was actually a prospecting expedition in order to establish the basis for a future expedition (Ligabue et al., 1972; Boccardi & Bottazzi, 1978). This expedition occurred the following year (November 4th–December 11th, 1973) and was an Italian-French expedition led by Giancarlo Ligabue and Philippe Taquet (Ligabue & Rossi-Osmida, 1975). A field report and a list of the excavated material was published in Ligabue & Rossi-Osmida (1975). The list included “1 [sic] Ouranosaurus nigeriensis” (p. 80). According to Rossi-Osmida (2005), all of the fossils collected during the Italian expedition were brought to the MNHM where they were prepared, restored and casted. In a letter dated August 27th 1974, Giancarlo Ligabue communicated to the Municipality of Venice his desire to donate a complete skeleton of an iguanodontian dinosaur and “other fossils found during the field campaign in the Sahara desert...undertaken in the years 1972/73”. The donation was accepted by the Consiglio Comunale (town council) of Venice on December 30th, 1974 (documentation is available at the MSNVE). In 1975, that skeleton, i.e., MSNVE 3714, was mounted in a room of the MSNVE and exhibited to the public along with the other specimens (including a complete skull of the crocodyliform Sarcosuchus imperator). Since that date, the skeleton has been on exhibition for the public in the museum.

In the formal description of the new species, Ouranosaurus nigeriensis, Taquet (1976, p. 58) mentions only the holotype (GDF 300), the paratype (GDF 381- MNHN) and the referred material (a large coracoid and a femur, indicated with the field acronyms GDF 301 and GDF 302, respectively). As stated above, no mention is made of the Ouranosaurus material supposedly collected by the 1973 Italian-French expedition in Taquet (1976). Despite being reported as a practically complete skeleton missing just the skull (p. 58), only the elements of the paratype that are not preserved in the holotype were described by Taquet (1976). A description of the whole paratype was never published. The holotype was returned to Niger after study (Taquet, 1976) and it is on exhibition at the MNBH in Niamey (Taquet, 1976, pl. IX, fig. 1). No further reference to the paratype, as well as the referred specimens GDF 301 and GDF 302, was made in the literature. According to A. McDonald (A McDonald, pers. comm., 2011) and Currie & Padian (1997, p. 369), the MNHM has only a plaster copy of the holotype.

Philippe Taquet (P Taquet, pers. comm., 2012) confirmed that the Venice specimen is the paratype with the missing bones being casts of the holotype. He also told us that he mapped the paratype bones in the field and that the map is still kept at the MNHN. Ronan Allain sent us a copy of that field map, which confirms that the Venice specimen contains the paratype material.

The field map of the Venice specimen

The field map sent to us by R Allain is divided into two sheets. The first sheet contains the writing “Ouranosaurus nig[eriensis]—Airfield—1970 —(specimen Venice Museum pro parte)” (in French) and the field map of the partially articulated paratype skeleton that is pictured in Taquet (1976, pl. IX, fig. 2; and also Taquet (1998), fig. 12) and was found at the margin of the landing strip built by the CEA in 1970. Furthermore, the map reports “GDF 381 today in Venice” (Fig. 2). The word “pro parte” (=for part) means that not all of the mapped bones were used in the mount of the Venice specimen or that the latter contains elements from other sources. Therefore, the correspondence of the bones reported in the map and those occurring in the mounted skeleton is checked below and the implications are discussed.

Figure 2 Ouranosaurus nigeriensis, the two sheets with the field maps drawn by P Taquet, redrawn and modified.

Part of sheet 1 with the writing “Ouranosaurus nig[eriensis]—Airfield—1970—(specimen Venice Museum pro parte)” was omitted. Some original handwritten notes have been translated into English and typewritten in dark gray. The author of the map marked with zig-zag lines some elements that he supposed to have wrongly drawn; those zig-zag lines are also in dark gray. When it is not possible to establish if the original identifications are correct, the names of the bones are in dark gray; black abbreviations are our identifications of the mapped bones or confirmed original identifications. “Near ulna” and “fragment of ulna” are a handwritten notes that refer to collected elements numbered 96 and 97, which were not drawn on the map. Abbreviations: ca, calcaneum; ch, chevron; co, coracoid; dv, dorsal vertebra; fe, femur; fi, fibula; h, humerus; il, ilium; mc, metacarpal; mt, metatarsal; ph, manus phalanx; pph, pedal phalanx; pu, pubis; ra, radius; sv, sacral vertebra; sc, scapula; st, sternal plate; ti, tibia; u, ulna. When the elements are reported as left in the original map, they are in brackets. See Supplemental Information 3 for further details.

The second sheet is the map of a set of bones, which is not clearly identifiable in Taquet (1976, pl. IX, fig. 2; also Taquet (1998), fig. 12). The relative location with respect to sheet 1 is unknown (see Supplemental Information 3 ). The morphology of the pelvic elements indicates that the partial skeleton belongs to a relatively large ornithopod; the shape of the pubes and the tall neural spines suggest that it belongs to O. nigeriensis.

The presence of a total of three pubes and possibly three ilia and scapulae, as well as a duplication of segments of the caudal vertebral column, indicates that the two sheets refer to two distinct skeletons.

Each bone in the two sheets is identified by a number in order to identify the elements and reassemble the skeleton once in the laboratory (Taquet, 1975); those numbers are not reported in Fig. 2, but are discussed in the Supplemental Information 3.

Systematic Palaeontology

Dinosauria Owen, 1842	
Ornithischia Seeley, 1887	
Ornithopoda Marsh, 1881	
Iguanodontia Dollo, 1888	
Ankylopollexia Sereno, 1986	
Styracosterna Sereno, 1986	
Ouranosaurus nigeriensis  Taquet, 1976	

Note: the name Ouranosaurus nigeriensis was first published by Taquet (1975, p. 41), without a formal description.

Holotype: GDF 300, a nearly complete skeleton, lacking the left maxilla, the right lacrimal, the right quadratojugal, the stapes, the articulars, dorsal vertebra 1 and probably another dorsal or two, the centrum of caudal vertebra 1 and caudals 25–26 and 30–31, most of the distal elements of the tail and some distal chevrons, one left metacarpal and most of the manus phalanges, both femora (only the distal condylar end of one of them was found), the left tibia, the left astragalus and calcaneum, the left metatarsals, and eight pedal phalanges. The skeletal elements in situ were scattered on a 15 m2 surface. The specimen is on exhibition at the Musée National Boubou-Hama in Niamey, Niger.

Paratype: GDF 381- MNHN (MSNVE 3714, “pro parte”, see below), partial skeleton without skull, but with the vertebral column in fairly good anatomical articulation and probably missing only the atlas and the distal segment of the tail.

Referred material: GDF 301, large coracoid; GDF 302, femur; and the elements of MSNVE 3714 that do not belong to the paratype (see below).

Horizon and Locality: Level GAD 5, upper part of the Elrhaz Formation, Tégama Series, Aptian, Aptian-Albian, or possibly Barremian, Early Cretaceous. All specimens are from the Gadoufaoua area of Niger. The holotype comes from the Camp des deux arbres locality, 7 km south east of Elrhaz, 16°42′ lat. N. 9°20′ long. E. The paratype was found 4 km south of the niveau des Innocents locality, along the eastern border of the airfield, 16°26′ lat. N, 09°08′ long. E. The exact locality for GDF 301 and GDF 302 was not reported in Taquet (1976).

Emended diagnosis: Styracosternan dinosaur with the following autapomorphies: thickened, paired domes on nasals, so that nasals extend further dorsally than frontals; maximum mediolateral width of the predentary over twice maximum rostrocaudal length along the lateral process; dorsoventral expansion of the anterior part of the dentary caused by the anterior divergence of the dorsal margin (the ventral margin is straight horizontal and the rostral end of the bone is not ventrally deflected); extremely tall neural spines in dorsal, sacral and proximal caudal vertebrae (up to seven times the height of the centrum in the middle dorsal vertebrae) forming a dorsal ‘sail’ with a sinusoidal outline (lower peak in the sacral segment); petaloid and flat brevis shelf in the ilium (without brevis fossa); U-shaped obturator gutter of ischium, much deeper than long; obturator opening of pubis bordered by the ischial peduncle and a ventromedial blade-like process starting from the basal part of the ischial peduncle (the opening is nearly encircled by the peduncle and process in medial view, while it appears as an obturator gutter in lateral view); distal extremity of the posterior ramus of pubis (pubis s. s.) slightly expanded and bulbous.

O. nigeriensis is also characterized by the following combination of characters that is apomorphic within the non-hadrosaurid styracosternans: elongate skull (length/height ratio =3.2) with laterally expanded and dorsoventrally flattened terminal part of the rostrum (“duck bill”) and oral margin of the premaxilla reflected dorsally to form a distinct rim (similar to some hadrosaurines); long ‘diastema’ in the dentary (as in Protohadros byrdi and hadrosaurids); tiny hand (humerus/metacarpal III length ratio >4 (similar to Uteodon aphanocetes and one specimen of Iguanodon bernissartensis) with spreading metacarpals.

Other potentially diagnostic characters include a circular orbit with the same height as the lower temporal fenestra; dorsal segment of the vertebral column made of only 15 vertebrae.

Notes on the diagnoses of Ouranosaurus nigeriensis

The original diagnosis by Taquet (1976, p. 60) is actually a summary of the overall anatomy of the species, not a list of apomorphies or an apomorphic combination of characters. At the time the diagnosis was written (over 40 years ago), only a few taxa were available for comparison (see Taquet, 1976 for a list of those taxa). Therefore, that diagnosis needed to be emended. Below is a detailed analysis of the purported diagnostic features of O. nigeriensis listed in the original diagnosis by Taquet (1976), in order to support their rejection or acceptance.

“Medium-sized iguanodontid (7 metres long)”—This cannot be accepted as a diagnostic feature. O. nigeriensis is not considered an iguanodontid (i.e., a member of the Family Iguanodontidae) anymore (see Sereno, 1986; Norman, 2004; McDonald et al., 2012b; Norman, 2015) and the actual length of a complete adult skeleton of this dinosaur taxon is unknown (see below). The ontogenetic stage of the holotype was not reliably established (see below); the paratype is an immature individual and is only slightly smaller than the holotype (see below). If the boundary between medium-sized and large-sized ornithopods is placed at 8 metres in length (Norman, 2015, p. 178), the holotype would probably approach it, if considering the complete tail in its body length estimate. That is the estimated body length of other styracosternans (e.g., Hypselospinus fittoni [7–8 m]; Dakotadon (=Iguanodon) lakotaensis [∼8 m]; Altirhinus kurzanovi [∼8 m]; and Eolambia caroljonesa [∼7–8 m]; Norman, 2015), so it cannot be apomorphic for O. nigeriensis. It would be the same even considering the present length (i.e., without the distal part of the tail) of the two known skeletons of O. nigeriensis because the body length of M. atherfieldensis and B. johnsoni is estimated at 6–7 m (Norman, 2015).

“Bipedal”—According to Norman (1980), the hind limb/forelimb length ratio and the index of forelimb proportions (radius/humerus length × metacarpal III/humerus length) provide information about quadrupedalism or bipedalism in a dinosaur. In MSNVE 3714, the hind limb/forelimb length ratio is 1.89, which is close to the values in M. atherfieldensis, Edmontosaurus annectens, and Lambeosaurus lambei (supposed to be bipedal) and it is unlike that of adult I. bernissartensis (which is supposed to be quadrupedal by Norman, 1980). The index of forelimb proportions is 0.14 in MSNVE 3714, which is closer to the values of I. bernissartensis (see Norman, 1980). Maidment & Barrett (2014) criticized the reliability of those ratios, identifying some osteological features that are correlated with quadrupedalism and that occur in O. nigeriensis: hoof-like unguals, straight femur that is longer than tibia, prominent and not-pendant fourth trochanter and pes/hind limb length ratio of 0.14. Bipedalism, facultative bipedalism or quadrupedalism would not be apomorphic for Ouranosaurus, in any case.

“Very long skull, narrow and relatively low, which maximum height occurs at the level of the nasal bulges”—This is a vague statement about elongation that should be supported by measurement ratios. The hadrosaurid E. annectens has a comparatively more elongate skull (length/height ratio is up to 3.4, while it is 3.2 in O. nigeriensis) and the basal hadrosauroids M. atherfieldensis and Tethyshadros insularis also have elongated skulls (ratios 2.6 and 2.57, respectively), although comparatively less than O. nigeriensis (see Dalla Vecchia, 2009). O. nigeriensis does indeed have the most elongated skull (length/height ratio >3) among the non-hadrosaurid styracosternans. The skull of O. nigeriensis is actually broader than the skull of M. atherfieldensis (compare them in dorsal view in Norman, 2004, fig. 19.4).

“Long and thin snout ending in a duck bill”—Proportionally, the snout of O. nigeriensis (the snout being the rostral part of the skull) is just slightly longer than that of M. atherfieldensis (rostrum/total skull length ratio is 0.66 and 0.60 respectively). “Duck bill” is a rather vague definition that is not adequately explained in Taquet (1976). The rostral expansion of the snout in dorsal view expressed as W/w (W = maximum skull width in dorsal view; w = maximum width of the snout) is 1.63 in O. nigeriensis (based on Taquet, 1976, fig. 10b) and 1.69 in M. atherfieldensis (based on Norman, 1986, fig. 4). However, the snout of O. nigeriensis is also quite flattened dorsoventrally unlike that of M. atherfieldensis and other non-hadrosaurid styracosternans. Furthermore, the anterolateral margin of the narial fossa above the occlusal edge of the premaxilla is reflected dorsally to form a distinct rim like in some hadrosaurines (Norman, 2002; Norman, 2015).

“Extremely long, straight and anteriorly expanded premaxillae, which separate posteriorly the nasals from the maxillae”—This is the condition observed also in M. atherfieldensis and in all hadrosauroids. The elongation of the premaxillae depends upon the elongation of the rostral part of the skull.

“External nares widely visible in dorsal view” and “orifice of convergence between the nasal ducts very back placed”—Taquet (1976, see fig. 16) considers as “external nares” the circumnarial depression (sensu Prieto-Márquez & Wagner, 2014) and as “orifice of confluence of the nares” the narial openings (“apertura ossis nasi” of Prieto-Márquez & Wagner, 2014, fig. 37.7). The circumnarial depressions are widely visible in dorsal view in other styracosternans where the skull can be observed in such a view, for example in M. atherfieldensis (see Norman, 1986, fig. 4) and in hadrosaurids (Horner, Weishampel & Forster, 2004). So, this is not an apomorphy of O. nigeriensis.

The external narial openings of O. nigeriensis are comparatively small and placed nearly at mid-rostrum. I. bernissartensis, M. atherfieldensis, Protohadros byrdi and B. johnsoni also have comparatively small external narial openings, which are in a slightly more anterior position in the rostrum than O. nigeriensis. However, this apparent posterior displacement of the openings in O. nigeriensis is only due to its more elongated rostrum.

“Short predentary bone, wider than long”—Many other styracosternans have predentaries that are wider than long (e.g., E. caroljonesa and all those listed in Prieto-Márquez, 2010, http://www.morphbank.net/Show/?id=461224, excluding Gryposaurus monumentensis). However, O. nigeriensis is the only iguanodontian to have a predentary maximum mediolateral width/maximum rostrocaudal length along the lateral process ratio (character 22 in Prieto-Márquez (2010); see http://www.morphbank.net/Show/?id=461224) that is higher than 2 (it is 2.35 based on measurements taken on Taquet, 1976, fig. 28).

“Low maxilla”—The maxilla of O. nigeriensis is not lower than those of many other styracosternans (e.g., M. atherfieldensis, E. caroljonesa and P. byrdi; see Gasulla et al., 2014, fig. 3). The maxilla appears to be low because of the elongation of the rostral part of the skull.

“Short nasal bearing a rounded dorsal bulge”—The “rounded dorsal bulge” on the nasal is indeed a unique feature of O. nigeriensis.

“Small antorbital fenestra”—The presence of a small antorbital fenestra is a primitive feature within the iguanodontians occurring for example in Tenontosaurus tilletti, Dysalotosaurus lettowvorbecki and Camptosaurus dispar (see Norman, 2004) and Hippodraco scutodens (see McDonald et al., 2010c). According to Norman (2015), O. nigeriensis shares with I. bernissartensis and M. atherfieldensis the antorbital fenestra perimeter, which forms a posteromedially directed canal when viewed laterally.

“Circular orbit with the same height as the lower temporal fenestra”—This is not due to a larger size of the orbit but to a comparatively small size of the lower temporal fenestra. In nearly all other styracosternans, the lower temporal fenestra is higher than the orbit (e.g., I. bernissartensis, M. atherfieldensis, Maiasaura peeblesorum and Prosaurolophus maximus), an exception being Parasaurolophus walkeri (see Horner, Weishampel & Forster, 2004, fig. 20.6B). However, the extent of the lower temporal fenestra is just hypothesized in many styracosternan species because skulls are incomplete, disarticulated or deformed by compression. Thus, the validity of the relative size of the two skull openings as a diagnostic feature needs to be confirmed.

“Straight and horizontal posterolateral process of squamosal that very little overlaps the paroccipital process”—This condition is found also in I. bernissartensis (see Norman, 1980, fig. 4B), M. atherfieldensis (see Norman, 1986, fig. 5) and P. gobiensis (see Norman, 2002, pag. 120).

“High paroccipital process, broad and anteriorly oblique”—The paroccipital process is similar in H. scutodens (see McDonald et al., 2010c, figs. 20 and 21); E. caroljonesa (see McDonald et al., 2012a, figs. 1 and 15), P. gobiensis (see Norman, 2002, fig. 3), B. johnsoni (see Godefroit et al., 1998, figs. 5 and 7) and Plesiohadros djadokhtaensis (see Tsogtbaatar et al., 2014, fig. 7.2)

“Broad and flat occipital condyle”—A relatively high morphological diversity exists among the occipital condyles of non-hadrosaurid styracosternans, but condyle shape was never considered as a diagnostic feature. The condyle is as broad as that of O. nigeriensis at least in E. caroljonesa (see McDonald et al., 2012a, fig. 18). It does not appear to be particularly flat in Taquet (1976, fig. 14) compared with those of I. bernissartensis (see Norman, 1980, fig. 9), Dakotadon lakotaensis (see Weishampel & Bjork, 1989, fig. 3) and Proa valdearinnoensis (see McDonald et al., 2012b, fig. 3A). Its articular surface is oriented caudoventrally, as in I. bernissartensis, M. atherfieldensis and Bolong yixianensis (see Wu & Godefroit, 2012).

“Basipterygoid processes directed mainly laterally”—This feature is observed also in I. bernissartensis (see Norman, 1980, fig. 4B), E. caroljonesa (see McDonald et al., 2012a, fig. 18E), Levnesovia transoxiana (see Sues & Averianov, 2009, fig. 1) and B. johnsoni (see Godefroit et al., 1998, figs. 6 and 7).

“Large upper temporal fenestrae, very divergent rostrally”—The upper temporal fenestrae are comparatively large and their main axis is laterocranially oriented also in M. atherfieldensis (see Norman, 1986, fig. 4), Jintasaurus meniscus (see You & Li, 2009, figs. 2A–B) and P. gobiensis (see Norman, 2002, fig. 8).

“Dentary that is deep anteriorly and low posteriorly”— E. caroljonesa (see McDonald et al., 2012a, figs. 3A–3B) and P. byrdi (see Head, 1998, figs. 11 and 14) also have a dentary that is deeper anteriorly than posteriorly. However, O. nigeriensis is unique in having a perfectly straight ventral margin of the dentary, i.e., there is no ventral deflection of the rostral end; the anterior increase of the dentary depth (up to the beginning of the tooth row) is due to the anterior divergence of the dorsal margin (see Taquet, 1976, figs. 29a, b and d).

“Dorsal margin of the dentary bearing a long diastema anteriorly”—According to Taquet (1976, p. 94), the “diastema” is the space between the first tooth and the posterior extremity of the predentary along the dorsal margin of the dentary. The extent of this “diastema” is variable among non-hadrosaurid styracosternans. It is very short in I. bernissartensis (see (Norman, 1980), fig. 2) and B. johnsoni (see Godefroit et al., 1998, fig. 5B), but it is longer in M. atherfieldensis (see Norman, 1986, fig. 3), A. kurzanovi (see Norman, 1998, fig. 3) and P. gobiensis (see Norman, 2002, fig. 3); it seems to be even longer in E. normani (see You et al., 2003, fig. 1a–c). The “diastema” of P. byrdi is only slightly shorter than that of O. nigeriensisis (see Head, 1998, figs. 11 and 14, p. 726). In hadrosaurids, the “diastema” is as long as that of O. nigeriensisis or longer (Horner, Weishampel & Forster, 2004).

“Well-developed retroarticular process [of the mandible]”—This process has the same development in other taxa, for example I. bernissartensis (see Norman, 1980), M. atherfieldensis (see Norman, 1986) and P. byrdi (see Norman, 2004).

“Teeth of Iguanodon-type, covered by enamel only on one side, with denticulated margins of the crown”—The sole fact of being like the teeth of “Iguanodon” discards this from being an apomorphy of O. nigeriensis. Teeth morphologically similar to those of O. nigeriensis occur in I. bernissartensis (see Norman, 1980), M. atherfieldensis (see Norman, 1986), A. kurzanovi (see Norman, 1998), Equijubus normani (see McDonald et al., 2014), Kukufeldia tilgatensis (see McDonald, Barrett & Chapman, 2010; McDonald et al., 2014) and Bolong yixianensis (see Wu & Godefroit, 2012). The enamel covering just one side of the crown is a diagnostic feature of Hadrosauromorpha according to Norman (2015, p. 176), who considers O. nigeriensis, I. bernissartensis, M. atherfieldensis and other taxa outside Hadrosauromorpha to have a thicker enamel layer on one surface of the crown and a thinner one on the other side (Norman, 2015, p. 186, character 57). Thus, Taquet (1976) could be wrong in considering Ouranosaurus teeth to be covered by enamel only on one side.

“Vertebral count: 11 [cervicals]—17 [dorsals]—6 [sacrals] 40 [caudals]” —The count of 11 cervicals is not diagnostic as it is the same in I. bernissartensis and M. atherfieldensis (see Norman, 2004), E. normani (see You et al., 2003), Xuwulong yueluni (see You, Li & Liu, 2011), B. yixianensis (see Wu & Godefroit, 2012) and T. insularis (see Dalla Vecchia, 2009). I. bernissartensis and M. atherfieldensis have 17 dorsals as well (Norman, 2004), while E. normani , Xuwulong yueluni and some hadrosaurids have 16 dorsals (You et al., 2003; Horner, Weishampel & Forster, 2004). However, if the dorsals are 15 in O. nigeriensis as seems probable, this would be an unusually low count, the same as Dryosaurus (Norman, 2004). Six is the plesiomorphic number of sacral vertebrae in basal hadrosauroids, occurring in M. atherfieldensis (see Norman, 2004), E. normani (see You et al., 2003), Xuwulong yueluni (see You, Li & Liu, 2011), P. gobiensis (see Norman, 2002), Gongpoquansaurus mazongshanensis (see Lü, 1997), Nanyangosaurus zhugeii (see Xu et al., 2000) and in the more basal Dryosaurus (Norman, 2004).

“Relatively short tail”—This apparent shortness is due to the fact that the caudal segment of the vertebral column is incomplete in both the holotype and the paratype. Consequently it is not a diagnostic feature.

“Extremely long neural spines of the dorsal vertebrae”—This is clearly a potentially diagnostic feature, but needs to be quantified because Hypacrosaurus altispinus, Barbsoldia sicinskii, Morelladon beltrani and GPIT 1802/1-7 (Iguanodontia indet.; Pereda-Suberbiola et al., 2011) also have tall neural spines on their dorsal vertebrae. The neural spines of the dorsal vertebrae of O. nigeriensis reach up to seven times the height of the centrum, while those of the other iguanodontians are shorter reaching less than five times the height of the centrum (see below). Furthermore, the neural spines of the sacral and proximal caudal vertebrae are also tall and altogether, they form a back ‘sail’, which has a sinusoidal outline unlike that of similar structures in other iguanodontians (see below). Neural spines of dorsal vertebrae flare apically in lateral view and the tallest spines have a paddle-like outline (“petal-shape” according to Pereda-Suberbiola et al., 2011, p. 557), with a basal neck and an expansion toward the apex, while those of other iguanodontians with tall neural spines have parallel or only slightly divergent margins (Maryanska & Osmólska, 1981; Pereda-Suberbiola et al., 2011; Gasulla et al., 2015).

“Long and straight ischium with a foot-like distal expansion”—The ischium of O. nigeriensis is not proportionally longer than that of I. bernissartensis (see Norman, 1980) or M. atherfieldensis (see Norman, 1986). The shaft in the Venice specimen is straight like that of the ischia of A. kurzanovi, Jinzhousaurus yangi and the hadrosaurid Shantungosaurus and Edmontosaurus, while it is slightly bowed in the holotype as in many other styracosternans (see Norman, 2015, character 95). A foot-like expansion at the end of the shaft of the ischium occurs in many styracosternans as well as in rhabdodontids and Camptosaurus dispar (see Norman, 2015, character 97).

“Very proximal obturator process” and “very narrow obturator gutter” [of ischium]—These are related features because it is the position of the process that makes the obturator gutter narrow. The obturator gutter is a broad embayment in I. bernissartensis (see Norman, 1980, fig. 67) and M. atherfieldensis (see Norman, 1986, fig. 53). It is narrower in other styracosternans like Hippodraco scutodens (see McDonald et al., 2010c, figs. 32a–b), P. gobiensis (see Norman, 2002, fig. 29), Gilmoreosaurus mongoliensis (see Brett-Surman & Wagner, 2006, fig. 8.6E) and some hadrosaurids (i.e., Sahaliyania elunchunorum and Nanningosauurus dashiensis; Godefroit et al., 2008, fig. 10B; Mo et al., 2007, fig. 1O), but it is always longer than deep. In B. johnsoni (see Godefroit et al., 1998, fig. 31) and some hadrosaurids (e.g., Hadrosaurus foulki, Maiasaura peeblesorum, Shantungosaurus giganteus and Kundurosaurus nagornyi; Prieto-Márquez, Weishampel & Horner, 2006, figs. 3C1–2; Guenther, 2014, fig. 22.1F; Brett-Surman & Wagner, 2006, fig. 8.6C; Godefroit, Bolotsky & Lauters, 2012, fig. 31) the obturator gutter is anteriorly bordered by a process that is distinct from the pubic process. The obturator gutter of O. nigeriensis is a U-shaped narrow slit that is much deeper than long (see Taquet, 1976, fig. 60), unlike all other styracosternans.

“Very elongated/slender [élancé ] pubis”—This is probably a mistake because the pubis of O. nigeriensis is less elongated/slender than that of I. bernissartensis (see Norman, 1980, figs. 64–65) and M. atherfieldensis (see Norman, 1986, fig. 55).

“Very deep and very developed prepubic blade”—Compared to the prepubic portion of the pubes of I. bernissartensis and M. atherfieldensis, the pubes of O. nigeriensis have a shorter and deeper neck. However, the prepubic blade is not much deeper than that of M. atherfieldensis and it is similar in depth and development to those of Lanzousaurus magnidens (see You, Ji & Li, 2005, fig. 3), Xuwulong yueluni (see You, Li & Liu, 2011, fig. 2), P. gobiensis (see Norman, 2002, fig. 28) and B. johnsoni (see Godefroit et al., 1998, fig. 32).

“Straight pubic rod, much shorter than ischium and with widened distal extremity”—The “pubic rod” (=posterior pubic ramus or pubis s.s.) is straight and shorter than the ischium in many styracosternans (Norman, 2015, character 94). However, the distal extremity is usually pointed (Norman, 2015), while it is slightly expanded and bulbous in O. nigeriensis. Taquet (1976) did not notice the peculiar morphology of the obturator opening of the pubis, which is unlike that of the other styracosternans (see below).

“Slender ilium”—The ilia of other styracosternans are similarly slender, for example those of I. bernissartensis (see Norman, 1980, fig. 63), Gilmoreosaurus mongoliensis (see Brett-Surman & Wagner, 2006, fig. 8.4G), Tanius sinensis (see Brett-Surman & Wagner, 2006, fig. 8.4E) and many hadrosaurids (Brett-Surman & Wagner, 2006, fig. 8.4A and C–D).

“Preacetabular process accounting for half the total length of the ilium”—The preacetabular process of O. nigeriensis is 47–50% the total length of the ilium. This is also the case of Planicoxa venenica (see DiCroce & Carpenter, 2001, fig. 13.5) and probably also of Iguanacolossus fortis (see McDonald et al., 2010c, fig.14). The preacetabular process is 47% of the total length in D. lettowvorbecki (see Galton, 1981, fig. 11I) and E. caroljonesa (see McDonald et al., 2012a, fig. 31).

“Convex dorsal margin of the ilium with a hint of an anti-trochanter”—In the Venice specimen the dorsal margin is actually straight, while it is convex in the holotype. Both conditions occur in a sample of I. bernissartensis, suggesting that the curvature of the dorsal margin is intraspecifically variable (Verdú et al., 2017). A “discrete bulbous boss present posterodorsal to the ischiadic peduncle” (Norman, 2015, character 90, p. 188) occurs also in P. valdearinnoensis, B. yixianensis, B. johnsoni and G. mongoliensis, as well as in Cedrorestes crichtoni (see McDonald et al., 2010c, fig. 18c).

“Shallow acetabular cavity”—The acetabular notch is not shallower in O. nigeriensis than in many other styracosternans, for example Altirhinus kurzanovi (see Norman, 1998, fig. 32), E. caroljonesa (see McDonald et al., 2012a, fig. 31) and P. gobiensis (see Norman, 2002, fig. 27).

“Shallow post-acetabular notch”—The notch is shallow also in Fukuisaurus tetorensis (see Carpenter & Ishida, 2010, fig. 2.7), NHMUK R3741 and NHMUK R9296 (Carpenter & Ishida, 2010, fig. 2.10a and b; the first referred to M. atherfieldensis by Norman, 2015) and in P. gobiensis (see Norman, 2002, fig. 27). The notch is not shallow in MSNVE 3714, suggesting a certain degree of individual variability (see below).

“Ascending process of the astragalus placed posteriorly instead of anteriorly”—Actually, Taquet (1976, p. 148) says that the astragalus of the holotype of O. nigeriensis has a posterior ascending process that is more developed than the anterior ascending process. This is a primitive feature occurring in Dysalotosaurus lettowvorbecki, as noticed by Taquet (1976) himself, and Eousdryosaurus nanohallucis (see Escaso et al., 2014). Tenontosaurus tilletti has no ascending processes at all (Forster, 1990), while most iguanodontians have a well-developed anterior ascending process (Norman, 2004). However, the astragalus of the Venice specimen differs from that of the holotype (see below), thus this feature cannot be considered a diagnostic feature.

“Tridactyl foot”—Actually, no complete foot of O. nigeriensis is preserved, so the presence of three toes is just assumed. Nevertheless, all styracosternans that have a preserved foot have three toes, so it would not be diagnostic of O. nigeriensis.

“Phalangeal formula [of the pes] 0-3-4-5-0”—This is just a hypothesis because no complete foot of O. nigeriensis is preserved. Nevertheless, it is the formula of all hadrosauroids with a preserved foot (Norman, 2004; Horner, Weishampel & Forster, 2004; Dalla Vecchia, 2009) and Dryosaurus (Norman, 2004), so it would not be diagnostic of O. nigeriensis.

“Humerus long and nearly straight”—The humerus/femur length ratio is 0.60 in O. nigeriensis (paratype). It is 0.56 in Uteodon aphanocetes (see Carpenter & Wilson, 2008), 0.83–0.62 in I. bernissartensis (see Norman, 1980), 0.57 in M. atherfieldensis (see Norman, 1986) and 0.62 in Jinzhousaurus yangi (see Wang et al., 2011). Thus, the humerus is comparatively not much longer in O. nigeriensis than in these taxa. It appears to be straight because of the low deltopectoral crest, which is a primitive feature within Styracosterna (Norman, 2004; Horner, Weishampel & Forster, 2004)

“Tiny hand bearing a spur-like and small fifth metacarpal that is not laterally directed”—Actually, the spur-like bone is not the metacarpal V but the ungual phalanx of manus digit I. According to Taquet (1976, p. 133) and the plaster copies mounted in the Venice specimen, the ungual phalanx of digit I of O. nigeriensis is “spur-like” as are those of I. bernissartensis, M. atherfieldensis and ‘Iguanodon mantelli’. According to McDonald et al. (2012a, p. 29), the ungual of manus digit I is a conical element in many basal styracosternans, including U. aphanocetes, Lurdusaurus arenatus, Barilium dawsoni, I. bernissartensis, M. atherfieldensis, O. nigeriensis, A. kurzanovi, J. yangi, E. caroljonesa and P. gobiensis. According to Norman (2015, character 80, p. 187), the ungual phalanx of manus digit I is a “conical spike” in C. dispar, an “enlarged and laterally compressed spine” in M. atherfieldensis, I. bernissartensis, O. nigeriensis, A. kurzanovi, J. yangi, H. fittoni, B. dawsoni and Bolong yixianensis, and a “small, narrow spine” in P. gobiensis. So, “spur-like” ungual phalanx of manus digit I is not an apomorphy of O. nigeriensis. As for size, the ungual phalanx of manus digit I of O. nigeriensis figured in Taquet (1976, fig. 53) is comparable to that of M. atherfieldensis (see Norman, 1986, fig. 50A). Furthermore, the size of that ungual phalanx can be intraspecifically variable (Verdú et al., 2017). The ungual phalanx of digit I is nearly perpendicular to the axis of the hand in I. bernissartensis (see Norman, 1980, figs. 60 and 62), M. atherfieldensis (see Norman, 1986, fig. 50A) and H. cf. fittoni (see Norman, 2015, fig. 38), but the direction of the phalanx could be the same in Uteodon aphanocetes (see Carpenter & Wilson, 2008, fig. 22A) and B. yixianensis (see Wu & Godefroit, 2012, fig. 19.9) as in O. nigeriensis. Furthermore, the condition in many other non-hadrosaurid styracosternans (e.g., A. kurzanovi, E. caroljonesa and P. gobiensis) is unknown because complete hands are rarely preserved. The non-perpendicular direction of metacarpal I and hence of the corresponding ungual phalanx with respect to the axis of the manus is a primitive feature, occurring in the basal ankylopollexian C. dispar (see Gilmore, 1909; Dodson, 1980, fig. 1D) and in the basal styracosternan U. aphanocetes. The hand of O. nigeriensis appears to be smaller than that of most styracosternans, but relative size is difficult to establish. Using the humerus/metacarpal III length ratio as a proxy of the relative manus size, that ratio is 4.4–4.5 in O. nigeriensis, 5.12 in U. aphanocetes (see Carpenter & Wilson, 2008) and 3.73–4.25 in I. bernissartensis (see Norman, 1980); it is lower than 4 in M. atherfieldensis (2.87; see Norman, 1986), J. yangi (3.75; see Wang et al., 2011), Nanyangosaurus zhugeii (2.3; see Xu et al., 2000), T. insularis (2.2; see Dalla Vecchia, 2009), the hadrosaurids Hypacrosaurus altispinus (2.18; see Brown, 1913) and Lambeosaurus magnicristatus (2.0; see Evans & Reisz, 2007). Similar results are obtained with the (humerus + ulna)/metacarpal III length ratio. The relatively small hand could be a primitive feature within iguanodontians because humerus/metacarpal III length ratio is 4.3 in Tenontosaurus tilletti (see Forster, 1990), 5.3 in Dysalotosaurus lettowvorbecki (see Galton, 1981) and 4.73 in Camptosaurus dispar (see Gilmore, 1909), and U. aphanocetes is a basal styracosternan in the phylogenetic analysis by McDonald et al. (2012b). However, caution is suggested in the absence of information about the ontogenetic stage of the sampled individuals because of the possibility of allometric growth during ontogeny. The metacarpals II–IV of O. nigeriensis (see below) are not compressed against each other to form a narrow and compact palm like in other styracosternans, for example I. bernissartensis (see Norman, 1980, figs. 60a–b and 61c–d), M. atherfieldensis (see Norman, 1986, figs. 51–52), A. kurzanovi (see Norman, 1998, p. 326) and P. gobiensis (see Norman, 2002, fig. 24). There are no articular facets for adjacent metacarpals III and IV or extended scars for ligamentous connections.

“Phalangeal formula [of the manus] 1-3-3-3-3 or 4”—This is just a hypothetical statement because no complete manus of O. nigeriensis is preserved. Nevertheless, it is the plesiomorphic formula for ankylopollexians and cannot be considered diagnostic of O. nigeriensis.

Description of MSNVE 3714 and Comparison with the Holotype

MSNVE 3714 is a partial skeleton; the missing elements were replaced by plaster copies, with the exception of the hyoid apparatus, the atlas and the cervical ribs, which are missing. Most of the original bones have also been partly reconstructed and restored (Fig. 3). Elements of the right side are more weathered then those of the left side because the paratype skeleton was exposed on the right side (as it is shown by the map; Fig. 2).

Figure 3 MSNVE 3714, Ouranosaurus nigeriensis, original and reconstructed parts in right (A) and left (B) views.

The reconstructed parts are in red.

In this section, only the skeletal elements of MSNVE 3714 that add new information with respect to the description of the osteology of O. nigeriensis by Taquet (1976) are described and compared with those preserved in the holotype.

Axial skeleton

The axial skeleton of MSNVE 3714 is composed of 76 vertebrae, but 10 caudals are completely reconstructed with plaster, so only 66 are actually preserved (Figs. 4A–4D). Curiously, 66 vertebrae are also preserved in the holotype, but the total count is 74 according to Taquet (1976) (Figs. 4E–4H; but see below). The axial skeleton of the paratype is a more reliable reference for the vertebral count in O. nigeriensis because it was in a better state of anatomical articulation with respect to the holotype (compare Fig. 2 and fig. 9 in Taquet, 1976).

Figure 4 MSNVE 3714 and holotype (GDF 300), vertebrae.

MSNVE 3714, the cervical series (A); the dorsal series (B); the sacrum (C); and the caudal series (D). Holotype, the cervical series (E); the dorsal series (F); the sacrum (G); and the caudal series (H). Numbers are progressive within each series. White vertebrae are those of the mount that are totally reconstructed. E–G are redrawn from Taquet (1976). Scale bar equals 50 cm.

Cervical vertebrae (Figs. 4A, 5A and 6). The axis and the following ten presacral vertebrae are preserved, while the atlas is missing. Presacral vertebra 11 has parapophyses that appear to be cut by the neurocentral suture (Fig. 5A), thus it is a cervical and not the first dorsal, according to the definition of the first dorsal vertebra by Norman (1986). Presacral vertebra 12 in the mounted skeleton has a relatively tall neural spine and its relatively small parapophyses are located just above the neurocentral suture (Fig. 5B), so it is the first dorsal vertebra. Therefore, MSNVE 3714 has 11 cervical vertebrae. Eleven cervical vertebrae are also preserved in the holotype (presacrals 1–11; Fig. 4E), but the neck was completely disarticulated in situ and presacral vertebrae 12–14 are not preserved, according to Taquet (1976). Eleven is also the cervical count of other styracosternans (Norman, 2004; Wang et al., 2011; McDonald et al., 2014). Therefore, a cervical count of 11 is supported for Ouranosaurus.

Figure 5 MSNVE 3714, cervical-dorsal transition.

The last (11) cervical (A); the first dorsal (B). Both are in left lateral view. Reconstructed parts are in dark gray colour. Abbreviations: dia, diapophysis; ncs, neurocentral suture; ns, neural spine; par, parapophysis; poz, postzygapophysis; prz, prezygapophysis. Scale bar equals 10 cm.

Figure 6 MSNVE 3714, cervical vertebrae.

Axis in cranial (A), right lateral (B), left lateral (C), and caudal view (D); cervical vertebra 3 in cranial (E), right lateral (F), left lateral (G), and caudal (H) views; cervical vertebra 11 in cranial (I), right lateral (J), left lateral (K), and caudal (L) views. Abbreviations: dia, diapophysis; od, odontoid process; ns, neural spine; par, parapophysis; poz, postzygapophysis; prz, prezygapophysis. Scale bar equals 10 cm.

The neural spine of the axis (Figs. 6A–6D) is low and sub-triangular in lateral outline, with a rounded dorsal margin. A broad circular depression occurs in the middle of both right and left sides of the spine. The neural spine in the axis of the holotype has a different M-like lateral outline (i.e., the dorsal margin is concave in the middle) and seems to lack lateral depressions (Taquet, 1976, fig. 37b). The other cervicals (Figs. 6E–6L) are similar to those of the holotype, as well as I. bernissartensis (see Norman, 1980) and many other styracosternans.

Dorsal vertebrae (Figs. 4B, 7 and 8). MSNVE 3714 has 17 dorsals (Fig. 4B). A series of 14 dorsals in relative anatomical connection is identifiable in the map of the in situ paratype (Fig. 2). A further centrum, which is displaced ventrally, may occur caudally to the last vertebra of the series. Thus, MSNVE 3714 has at least two dorsal vertebrae more than the paratype. Possibly, two additional distal-most vertebrae could have been covered in the field by a broad bone present in the corresponding area (plausibly an ilium) and were not mapped. However, this seems to be unlikely because those vertebrae would overlap the sacrum (Fig. 2). More plausibly, two vertebrae from another specimen were added to the paratype material to complete the mounted skeleton because the holotype was supposed to have 17 dorsals (Fig. 4F).

Figure 7 MSNVE 3714, dorsal vertebrae in lateral view.

(A) left side; (B) right side. The parts of the neural spines that have been reconstructed or just covered by resin are highlighted in white. Reconstructed parts of the centrum, transverse processes, zygapophyses and pedicels of the neural arch are not highlighted. Numbers are progressive. Scale bar equals 10 cm.

Figure 8 MSNVE 3714, dorsal vertebrae.

Dorsal vertebra 1 in right lateral (A), caudal (B), and ventral view (C); dorsal vertebra 9 in left lateral (D), cranial (E), and ventral view (F) views; dorsal vertebra 17 in left lateral (G), caudal (H) and ventral (I) views. Abbreviations: bpl, ‘bump’ of the prespinal lamina; kl, keel; par, parapophysis. Scale bars equal 10 cm.

The peduncles of the neural arches, parapophyses, transverse processes and relative diapophyses are all reconstructed in the Venice specimen, presumably taking as reference for proportions and morphology those of the holotype. The neural spines, which are the most important feature of those skeletal elements, are also partly restored (Fig. 7).

The centra of dorsals 1 and 2 are opisthocoelous. From dorsal vertebra 3 onwards, they become slightly amphicoelous to amphiplatyan. Their length ranges from 87 (vertebra 12) to 112 mm; they are slightly longer than high but dorsal 17 is more elongated than the others; Figs. 8G–8I). The centrum of dorsal 1 has a ventral longitudinal keel and sub-circular articular surfaces like those of the cervicals (Figs. 8A–8C) and is smaller than the centrum of the last cervical (Fig. 5). In ventral view, all other dorsal centra are spool-shaped with a keeled ventral margin (Figs. 8C and 8F); only centrum 17 apparently lacks a keel (Fig. 8I). Dorsal 17 could actually be a dorsosacral because the sacral vertebrae have a faint ventral keel or lack this feature. The articular facets of centra 2–16 are higher than wide, while the reverse is true in centrum 17.

The morphology of the tall neural spines changes along the vertebral column (Fig. 7). The spine of dorsal 1 is straight and inclined caudally (60°); it slightly tapers apically in its basal part, while the apical half has parallel caudocranial margins and does not flare apically (Figs. 5B and 8A–8B). The spine is only 1.41 times the height of its centrum. The spine of dorsal 2 is incomplete apically. The preserved part is 2.7 times the height of the centrum. As reconstructed, it is about four times the height of the centrum. It is narrow craniocaudally, slightly sloping caudally (about 80°) and slightly arched; it was probably not expanded apically because the cranial and caudal margins are parallel (thus, the reconstruction is correct). The spine of dorsal 3 is taller and craniocaudally longer than that of dorsal 2, but it is also incomplete apically. It is straight, with nearly parallel craniocaudal margins and nearly oriented vertically (about 83°). The apex and part of the apical tract of the caudal margin of the spine are reconstructed, but the spine was probably not greatly expanded craniocaudally. Dorsal 4 has a neural spine that is taller and craniocaudally longer than that of the preceding vertebra, but it is incomplete apically as well. The spine is straight and reaches its minimum craniocaudal length just below the mid-shaft; its cranial and caudal margins diverge above the point of minimum craniocaudal length, so the apex was probably slightly expanded. Unlike the preceding vertebra, the spine slopes cranially (about 5°from the vertical). Dorsal 5 has a spine that is nearly complete apically and is taller than that of the preceding vertebra. The spine is straight and slightly sloping cranially. Minimum craniocaudal length occurs in its lower third, but it is unclear whether this is a real feature or an artifact of preparation. The cranial and caudal margins diverge above that point, so the spine much flares toward its apex, which is rounded and asymmetrical. The spine of dorsal 6 is straight and vertical. It is incomplete apically; nevertheless, it is at least as tall as the spine of the preceding vertebra. The cranial and caudal margins diverge above the lower third, so the spine much flares toward the apex. Unlike that of the preceding vertebra, the spine of dorsal 7 is recurved cranially. Similar to spine 6, it flares apically. As the apical portion is partly reconstructed, its squared outline is hypothetical. The cranial curvature cannot be a real feature because it would prevent zygapophyseal and central articulation with the preceding vertebra, unless the spines of the two vertebrae overlapped laterally. The spine of dorsal 8 is unlike those of the preceding and following vertebrae. It appears to be craniocaudally narrower, sloping caudally and flaring above its basal third. The apex is reconstructed, so its squared outline is hypothetical and its total height is unknown. The spine of dorsal 9 is slightly curved at the base, but the rest is straight and vertical (Figs. 8D–8E). Its basal portion is reconstructed; if the reconstruction is correct, this spine is the tallest (it is seven times the height of its centrum). Flaring starts in the basal part of the spine. The apex is partly reconstructed, so its squared outline is hypothetical. The spine of dorsal 10 is also arched basally and flares starting from the basal portion; its apical third is mostly reconstructed, so nothing can be said about its real outline. The whole spine of vertebra 11 seems to be slightly recurved, but its basal part is reconstructed, so this feature could be an artifact. The preserved portion of the apical part shows that this spine was shorter than spine 9. Spine 12 seems to be arched basally and straight from mid-shaft onwards. Its apical portion is mostly reconstructed, so its real height and the shape of its apex are unknown. The following neural spines 13–17 are all arched (less in the spine 15, which is poorly preserved) and flare apically like the preceding ones, although proportionally less than in middle dorsals (Figs. 8G–8H). Their craniocaudal length decreases slightly moving caudally. Their height decreases markedly; spine 14 is just slightly lower than spine 11, but the decrease is marked in the following vertebrae 15–17.

The basal part of the spine in vertebrae 5, 7, 8, and 16–17 shows a cranial bump that is made by the cranially expanded prespinal lamina (Figs. 7 and 8G) and is observed also in the distal dorsals of the holotype (Fig. 4F). All neural spines are laterally flattened and they do not thicken apically.

According to Taquet (1976, p. 109), the holotype preserves 13 dorsals but the cervical series was separated in situ from the first preserved dorsal vertebra by a gap that could be filled by other dorsal vertebrae or just be caused by displacement of the dorsal and cervical segments. Taquet (1976) opts for the first hypothesis, suggesting that the first four dorsals were missing. However, only the first three dorsal vertebrae (the presacral vertebrae 12–14) are missing in the reconstruction of the vertebral column by Taquet (1976, fig. 38; here Fig. 4F). Furthermore, Taquet (1976, p. 109) says that the dorsal vertebrae following the cervical-dorsal gap were not scattered in the field (i.e., they were in anatomical connection), but fig. 9 of Taquet (1976) shows that this is the case only for a segment of just nine dorsal vertebrae. So, it is unclear how the count of four missing dorsal vertebrae and the total count of 17 dorsal vertebrae were established (see Taquet, 1976; figs. 38 and 40). The much better articulated vertebral column of the paratype shows that Taquet (1976) is wrong in his reconstruction of the holotype dorsal vertebral series. Only one of the supposedly missing proximal dorsals of the holotype (see Fig. 4F) is present in MSNVE 3714; it corresponds to the first dorsal. The second dorsal of MSNVE 3714 corresponds to vertebra 4 of the holotype (see Fig. 4F). Dorsal vertebra 5 of the holotype has a cranially sloping neural spine that would cause the crossing with the neural spine of the preceding vertebra when the two vertebrae are in anatomical articulation; furthermore, the spine tapers apically. There is no such vertebra in MSNVE 3714: dorsal 3 is morphologically similar to vertebra 6 of the holotype (compare Figs. 4B and 4F). Dorsal vertebra 4 of MSNVE 3714 corresponds to dorsal 7 of the holotype in the relative height and slight cranial slope of the neural spine, although the spine of the holotype is paddle-like. In both skeletons, the first middle dorsal vertebrae tend to have straight vertical neural spines that are craniocaudally expanded apically, while last middle dorsal and distal dorsal vertebrae have arched spines whose craniocaudal expansion decreases posteriorly. However, the number of middle and distal elements is different in the two specimens. It is evident that the spines of dorsals 7 and 8 are unlike from those of the contiguous vertebrae in MSNVE 3714 (Fig. 7). This suggests that those two vertebrae were added to maintain the estimated count of 17 vertebrae reported for the holotype (see the Discussion). This is supported by the lower vertebral count in the field map of the paratype. The comparison between the holotype, the field map of the paratype and MSNVE 3714 suggests that O. nigeriensis had a shorter torso (possibly with 14 dorsals and one dorsosacral) and that the tallest neural spine is that of dorsal vertebra 7 (9 in Figs. 4B and 4F). In the only photo of the holotype exhibited at the MNBH available in the internet (http://www.gettyimages.it/detail/fotografie-di-cronaca/herbivorous-dinosaur-skeleton-of-ouranosaurus-fotografie-di-cronaca/543868764#herbivorous-dinosaur-skeleton-of-ouranosaurus-nigeriensis-taqueti-picture-id543868764), the dorsal segment of the vertebral column is composed of only 13 articulated vertebrae.

Some other inconsistencies regarding the dorsal vertebrae are found in Taquet (1976). Dorsal vertebrae 10–12 are reported to have the highest neural spines (p. 112), but the tallest is actually the spine of the ninth dorsal of fig. 38, while height decreases gradually in the following vertebrae, which is confirmed by the measurements reported on pages 178–179. As reported above, dorsal vertebra 9 is the sixth preserved vertebra in the holotype and it would be dorsal 7, if only the first is missing of the preceding dorsal vertebrae as suggested by MSNVE 3714. According to Taquet (1976, p. 112), the highest neural spine is 3.9 times the height of its centrum, but it is actually nearly seven times according to fig. 38 (7.11 according to the measurements on p. 178). The centrum of the dorsal vertebra with the highest spine is reported to be 160 mm high, but it is actually less than 90 mm high in dorsal 8, according to the scale bar in fig. 41, and that of dorsal 9 is 90 mm according to measurements on p. 178.

Sacrum (Fig. 9). The sacrum of MSNVE 3714 is composed of six fused vertebrae (Figs. 4E, and 9) like that of the holotype. A shallow longitudinal keel extends along the ventral surface of sacral centra 1–2 and becomes very faint in centra 3–4 and is not evident in centra 5–6. Centrum 6 has a nearly flat ventral side. The neural spines are straight, vertical and only slightly craniocaudally longer in the apical part than in the basal part. The spines of sacral vertebrae 1–3 are of similar height; they increase in height from sacral 4 up to 6, which bears the tallest spine. Therefore, the last two sacral spines form the beginning of the caudal hump of the ‘sail’. The spines are regularly separated, except the last one: the distance between spines 5 and 6 is twice the distance between spines 4 and 5. Apparently, the trend in neural spine height is the reverse in the holotype: height decreases from sacral 1 to sacral 5 (the spine of sacral 6 is not preserved). Thus, there is a step in the ‘sail’ outline in correspondence of the passage between the dorsal vertebrae and the sacrum as well as at the passage between sacrum and caudal vertebrae (Figs. 4F–4H). This condition is probably artificial and MSNVE 3714, showing a more gradual transition from the dorsal to the sacral and from the sacral to the caudal spines (Figs. 1 and 4B–4D), appears to be more reliable.

Figure 9 MSNVE 3714, the sacrum.

Left lateral view. Scale bars equal 10 cm.

Caudal vertebrae (Figs. 4D) and (10A–10K). The tail is composed of 43 caudal vertebrae, but five vertebrae (caudals 27–31) and the terminal string of five vertebrae (caudals 39–43) are made of plaster (Fig. 4D). Thus, there are 33 original vertebrae. The total caudal count was surely higher (see the caudal count of several dinosaur taxa in Hone, 2012), possibly higher than the count in Iguanodon bernissartensis (46), whose tail is distally incomplete (Norman, 1980) or even much higher (the count is over 75 in TMP 98.58.01, an indeterminate hadrosaurid; FM Dalla Vecchia & M Fabbri, pers. obs., 2011).

Figure 10 MSNVE 3714, caudal vertebrae.

Vertebra 6 in caudal (A) and right lateral (B) views; vertebra 10 in cranial (C), right lateral (D), left lateral (E) and ventral (F) views; vertebra 21 in left lateral (G), caudal (H) and ventral (I) views; vertebra 35 in left lateral (J) and caudal (K) views; haemapophysis 8 in left lateral (L) and cranial (M) views. Abbreviations: af, articular facets of the haemapophysis, afh, articular facet for the haemapophysis; bpl, bump of the prespinal lamina; ns, neural spine; pla, pleurapophysis; poz, postzygapophysis; prz, prezygapophysis; vld, ventral longitudinal depression of the centrum. Scale bar equals 10 cm.

There are 20 proximal and 12 middle caudal vertebrae (17 including the five that are totally reconstructed). The last preserved caudal (caudal 38) seems to be a distal element (but see below).

The holotype also preserves 33 caudals. According to Taquet (1976, p. 118) four further vertebrae are missing from the tail of the holotype (two between caudals 24 and 27 and two between caudals 29 and 32; Fig. 4H). However, this reconstruction is hypothetical because that tail was partly disarticulated (Taquet, 1976, fig. 9). The holotype has 14–15 proximal caudals (15 in the text, 14 in fig. 44) and more than 12 middle caudals (at least 14 considering the two hypothetical ones after caudal 24). The last nine caudals should be distal elements (Taquet, 1976, fig. 44) since the haemapophyseal facets can be observed up to the posterior portion of caudal 31, according to the text (p. 119). However, caudals 30 and 31 are not preserved according to fig. 44, thus the actual number of mid- and distal caudals is uncertain in the holotype.

I. bernissartensis has 14 proximal, about 22–24 middle and at least 8–10 distal caudal vertebrae (Norman, 1980). Mantellisaurus atherfieldensis (IRSNB 1551; Norman, 1986) has 15 proximal and at least 17 middle caudals (the tail is incomplete distally). So, MSNVE 3714 has five and four proximal caudals more than I. bernissartensis and M. atherfieldensis, respectively. This suggests the presence of a M. caudifemoralis that extended more caudally in the African taxon than in the European ones (Persons & Currie, 2011). On the other hand, MSNVE 3714 seems to have a comparatively low mid-caudal count. However, its only distal caudal element is possibly the vertebra without number that was not mapped and was probably not closely associated with the others (see Fig. 2); therefore, it could be the only collected distal element, which was later attached to the last preserved middle caudal. Another possibility is that vertebra 38 is just one of the last middle caudals and the haemapophyseal facets were weathered away. In both cases, the actual middle caudal count of the paratype would be higher. The middle caudal counts of the Venice specimen, I. bernissartensis and M. atherfieldensis suggest that the count hypothesized by Taquet (1976) in the holotype is too low.

In MSNVE 3714, centra are slightly amphicoelous in the proximal and first middle caudals to become amphiplatyan caudally. The caudal surface of proximal and middle caudals is more squared than the rounded cranial one because of the presence of the raised facets for the haemapophysis (Figs. 10A, 10C and 10H). The latter appear in the third caudal; however, articular facets for the haemapophysis occur also cranially in caudals 3–10. Centra are constricted in the middle and are hourglass-shaped; they are shorter than tall up to vertebra 17 and shorter than broad up to vertebra 19 (Table S1). The centrum of caudals 1 and 2 has a longitudinal ventral keel, which is faintly developed in caudal vertebra 3. The following centra 4–10 seem to have a convex ventral face, but they are poorly preserved. From vertebra 11 up to vertebra 33, the ventral side of the centrum has a broad longitudinal depression (probably a haemal groove), which seems to correspond with the shift of the haemapophyseal facets on to the caudal part of the centrum only.

The lateral surface of the centrum near its articular facets is rough, with longitudinal grooves in some proximal and all middle caudal vertebrae (caudals 11–25; Fig. 11A), suggesting the presence of a cap of cartilage. The neurocentral suture is still visible in proximal and middle caudal vertebrae (Figs. 11B–11D) up to caudal 25.

Figure 11 Evidence of osteological immaturity in the caudal vertebrae of MSNVE 3714.

Rough surface in vertebral centrum 24 (A); neurocentral suture in vertebra 8 (B); vertebra 10 (C); and vertebra 21 (D). Vertebrae are figured in left lateral view. Arrows point to the grooved surface in A and to the neurocentral suture in B–D. Scale bar equals 5 cm in A and 10 cm in B–D.

The pleurapophyses of the proximal caudals are flattened dorsoventrally and scarcely project laterally (Fig. 12). They occur at the base of the neural arch on a ventral expansion of the pedicel overlapping the centrum laterally. They decrease in size along the series becoming knob-like; they disappear totally in caudal vertebra 21 (Fig. 12C), but in caudals 19 and 20 they are just small bumps (Fig. 12B).

Figure 12 Proximal to middle caudal transition in MSNVE 3714.

Vertebra 18 (A); vertebra 20 (B); vertebra 21 (C). They are shown in dorsal view. Abbreviations: ns, neural spine; pla, pleurapophysis; prz, prezygapophysis. Scale bar equals 5 cm.

The neural spines are mostly spatulate in lateral view, with a slight craniocaudal apical expansion (Figs. 4D and 10). They are inclined caudally to different degrees. For example, the spine of caudal vertebra 2 is only slightly sloping (77.2°), while those of vertebrae 7, 9 and 16 slope 52.7°, 58.4°and 48.6°, respectively (Fig. 4D). The proximal spines are mostly straight (Fig. 10B), but spines of vertebrae 6, 10 (Figs. 10D–10E) -12, 14–15 and those posterior to vertebra 17 (Figs. 10G–10H) are arched with a cranial-facing concavity (Fig. 4D). Non-harmonic sloping of the spines along the vertebral column is probably a restoration bias because the proximal portions of some neural spines were broken into several pieces that have been glued together and missing portions have been reconstructed. The basal arching of spines 3 and 4 (which does not occur in preceding and following spines; Fig. 4D) could also be a consequence of restoration. Neural spine inclination and morphology in the caudals of the holotype are more regular than in MSNVE 3714 (Taquet, 1976, figs. 40 and 43–44; Fig. 4H). Additionally, the spines of vertebrae 1–4 are slightly arched backward in the holotype (Fig. 4H), unlike those of the Venice specimen. Proximal caudals 2–7 present a cranially projecting bump of the basal part of the prespinal lamina that occurs only in caudals 1–3 of the holotype.

Haemapophyses (Figs. 10L–10M). There are 26 haemapophyses (chevrons), but seven are completely reconstructed. Haemapophyses 1–18, 20, and 23 are original, although they all contain reconstructed parts; chevrons 19, 21–22 and 24 to the last one are all artificial.

The first haemapophysis is located between caudals 3 and 4, while the last one occurs between vertebrae 28 and 29, but it is artificial like the two vertebrae. There are two articular facets per pedicel in the chevrons of the first proximal caudal vertebrae because each pedicel articulates on two centra. The dorsoventral length of the haemapophyses tends to decrease caudally, but chevron 14 is shorter than chevron 15, both preserved distally. Possibly chevron 14 is in the wrong position and should be placed in a more distal position. The spine of haemapophysis 1 is straight, but those of the following elements up to haemapophysis 5 are arched, while the following show a variable degree of curvature from nearly straight to slightly arched. As with the caudal neural spines, the morphology and sloping of the chevrons of the Venice specimen are less regular and harmonic than in the holotype. This is probably a consequence of the breakage of the long and thin spines and subsequent restoration.

In both basal and derived iguanodontians, chevrons 1–3 (up to chevron 6 in some taxa) taper distally (e.g., Tenontosaurus tilletti, Forster, 1990, fig. 5A; Iguanodon bernissartensis, Norman, 1980, fig. 47; Mantellisaurus atherfieldensis, Norman, 1986, fig. 39; Xuwulong yueluni, You, Li & Liu, 2011, fig. 2; Tethyshadros insularis, Dalla Vecchia, 2009, fig. 1; Kritosaurus incurvimanus, Parks, 1920, pl. 1; Brachylophosaurus canadensis, Prieto-Márquez, 2001, fig. 52; Corythosaurus casuarius, Brown, 1916, fig. 2). Furthermore, those chevrons are more inclined caudally than the following chevrons and they touch each other (see I. bernissartensis, Norman, 1980; X. yueluni, You, Li & Liu, 2011; T. insularis, Dalla Vecchia, 2009; K. incurvimanus, Parks, 1920; B. canadensis, Prieto-Márquez, 2001). This is not the case for both the holotype and MSNVE 3714 (see Fig. 1 and Fig. 4H), suggesting that the tails of those skeletons were reconstructed and mounted incorrectly.

Ossified tendons. Unlike other iguanodontians, no ossified tendons are preserved with MSNVE 3714. According to Taquet (1976, p. 113), they were represented by a few fragmentary remains in the holotype and possibly by their traces on the neural spines of the distal dorsals. While the rarity of ossified tendon remains in the holotype could be caused by taphonomic reasons (the vertebral column is disarticulated), their total absence in the articulated vertebral column of the paratype is puzzling. Possibly, the characteristic lattice that occurs laterally on the dorsal to proximal caudal vertebrae of the iguanodontians was scarcely developed in this high-spined taxon (contra Organ, 2006b).

Appendicular skeleton

Coracoid (Figs. 13A–13C). The right coracoid is a plaster copy, while the left one is original and smaller than the right one (it is about 8% shorter). Taquet (1976) described the right coracoid of the paratype instead of that from the holotype because of the bad preservation of the latter. The right coracoid of the Venice specimen is a plaster replica of the right coracoid of the paratype described and figured by Taquet (1976, p. 124, fig. 48). Unlike MSNVE 3714, the coracoids of the holotype were fused to their scapulae.

Figure 13 MSNVE 3714, pectoral girdle elements.

Left coracoid in caudal (A) and dorsomedial (B) views; drawings of the left coracoid in caudal and dorsomedial views with the reconstructed parts evidenced in dark gray colour (C). Left scapula in cranial (D) and lateral (E) views; drawing of the left scapula in lateral view with the reconstructed parts evidenced in dark gray colour (F). Left sternal plate in dorsomedial (G) and ventrolateral (H) views; drawings of the left sternal plate in dorsomedial and ventrolateral views with the reconstructed parts evidenced in dark gray colour (I). Abbreviations: ap, acromion process; cof, coracoid foramen; clp, caudolateral process (‘handle’); crmp, craniomedial plate; cvp, caudoventral process; cos, coracoid sutural surface; df, deltoid fossa; dr, deltoid ridge; gl, glenoid; scl, scapular labrum; scs, scapular sutural surface; sp, broken sternal process. Scale bar equals 10 cm.

Scapula (Figs. 13D–13F). Both scapulae are preserved, but the blade of the right one is mostly reconstructed. The left scapula (640 mm long) is 3% longer than the scapula from the holotype. The left scapula shows that the blade expands distally into a symmetrical spatula (the distal portion is not completely preserved in the holotype).

Sternals (Figs. 13G–13I). Both sternals are preserved in MSNVE 3714, although they are partly reconstructed (Fig. 13I). They are hatchet-shaped with an expanded and broad proximomedial portion (the sternal ‘paddle’) and a rod-like caudolateral process (the sternal ‘handle’). Their maximum length is 330 mm (6% longer than the sternals of the holotype). The ‘paddle’ has a triangular caudoventral process that is not preserved in the sternals of the holotype. The ‘handle’ is slightly expanded and thickened to the tip.

Humerus (Figs. 14A–14D). Both humeri are original in MSNVE 3714 and are 510 mm long (92% the length of the humerus of the holotype). They are slightly more robust than those of the holotype (total humeral length/minimum shaft width ratios are 8.51 and 10.45, respectively). The deltopectoral crest is much longer than wide (i.e., it is scarcely prominent) and symmetrical (the apical point is in the middle of the crest and there is no steep distal margin). Its blunt apex is in the proximal half of the humerus. The crest is slightly more prominent than that of the holotype (total humeral length/width at level of the apex of the deltopectoral crest ratios are 5.56 and 7.50, respectively). The ulnar condyle is distally projected further than is the radial condyle with respect to the ulnar condyle of the holotype. This intraspecific variability in the distal (ventral) projection of the ulnar condyle was reported also in I. bernissartensis (see Verdú et al., 2017).

Figure 14 MSNVE 3714, forelimb, long bones.

Left humerus in caudal (A), cranial (B), proximal (C) and distal (D) views; left radius-ulna in caudal (E), cranial (F), medial (G), lateral (H), proximal (I) and distal (J) views. Abbreviations: ch, caput humeri; dpc, deltopectoral crest; gtb, greater tuberosity; itb, inner tuberosity; lf, lateral flange; mf, medial flange; ol, olecranon; ra, radius; rc, radial condyle; u, ulna; uc, ulnar condyle. Scale bar equals 10 cm.

Ulna and radius (Figs. 14E–14J). Both radius-ulna pairs are preserved; the elements of each pair are stuck together by glue. The ulna is 88% the length of the ulna of the holotype and its robust olecranon is larger and better developed than that of the ulna of the holotype (see Taquet, 1976, fig. 51a, c). The radius is 84% the length of the radius of the holotype.

Carpals(Fig. 15A). The right carpus is artificial and made of replicas of the left carpals. The left carpus is made of a proximal row of three bones, which are glued to their relative metacarpals (which, however, are artificial). One carpal apparently articulates with phalanx I-1 and metacarpal II, therefore it is in the position of the radiale. The middle element articulates with metacarpal III and possibly, partly with metacarpal IV and can be identified as the intermedium. The third carpal articulates with metacarpal IV and should be the ulnar. The ‘radiale’ is the smallest of the three; it has a blocky shape, with a sub-quadrangular outline, an irregular and probably damaged proximal surface and a slightly concave and smooth distal surface. The ‘intermedium’ is the largest element and has a quadrangular outline in proximal and lateromedial views (it is dorsoventrally higher than lateromedially wide). Probably it is made of two cubic bones glued together. The ‘ulnare’ is a blocky cylindrical element that is proximodistally longer than ventrodorsally high in lateromedial view. The wrist of the holotype is completely different, being formed only by two large and proximodistally flattened proximal elements. According to Taquet (1976, p. 129), the largest element articulated with the radius and is composed of the radiale fused to the carpals I, III and IV and to metacarpal I, while the other element is made of the fused intermedium, ulnare and carpal V. Since the larger element was found still articulated to the right radius, its identification and location is not a matter of interpretation. The wrists of Iguanodon bernissartensis (see Norman, 1980, fig. 60) and Mantellisaurus atherfieldensis (see Norman, 1986, fig. 50) are formed of relatively small-sized and block-like carpals, which are fused into a single element in I. bernissartensis, while they are partly separated in M. atherfieldensis. Unlike the carpus of MSNVE 3714, the largest element in both taxa is the radiale (which is fused to metacarpal I, as usual in non-hadrosauromorph ankylopollexians; Norman, 1980, figs. 59–60; Norman, 2015, character 79 and relevant codings on p. 98) and two to five distal carpals can be identified. We suspect that those mounted in MSNVE 3714 are not actually carpals or at least they do not preserve their original shape.

Figure 15 MSNVE 3714, forelimb, manus in dorsal (cranial) view.

Left (A) and right manus (B). (C) and (D) are the drawings of the left and right manus, respectively. Totally reconstructed parts are evidenced in dark gray colour and those just covered by a film of resin are pale gray coloured; also minor reconstructed portions are pale gray. The reconstruction of the articulated digits II–V is reported in E. Abbreviations: int, intermedium; mc II–V, metacarpals II–V; mk, medial distal knob on the metacarpal V; ph II–V, phalanges of manus digits II–V (the last phalanx of each digit is the ungual); rad, radiale; uln, ulnare. Elements without abbreviation in A and B are reconstructed. Scale bar of A and B equals 10 cm.

Metacarpals (Fig. 15). Metacarpals from the left manus are artificial, whereas those of the right one are original with minor restoration (Figs. 15C and 15D). Metacarpals III and IV are the longest, metacarpal III being only slightly longer than metacarpal IV (113 and 108 mm, respectively). The shortest is metacarpal V (68 mm long). The right metacarpals of the paratype are figured in Taquet (1976, fig. 57a); with respect to drawings in that figure, the metacarpals II–IV of MSNVE 3714 are slightly narrower at mid-shaft and hence slenderer.

The orientation of the manus in the following description (that integrates the description by Taquet, 1976) is the standard one for digitigrade quadrupedal tetrapods, although Ouranosaurus probably kept the palms facing somewhat medially.

Metacarpal II (85 mm long) is straight, relatively slender (it is constricted in the middle) and with expanded extremities. Its proximal end is expanded craniopalmarily and mediolaterally. Its proximal surface is flat (it is convex in Taquet, 1976). The lateral side is flat and rough up to mid-shaft. The palmar side is flat too. The distal surface has a sub-oval outline and is concave. As underlined by Taquet (1976, p. 131), metacarpal II from the holotype is slenderer and mediolaterally flattened than this metacarpal.

Metacarpal III is expanded mediolaterally at both extremities and has a straight shaft. Its proximal surface is gently convex (it is strongly convex for Taquet, 1976). The proximal part of the medial side has a flat facet where the corresponding flat side of metacarpal II abuts it. So, metacarpal II is wrongly displaced distally in the mounted hand. The lateral side of the shaft has a thin longitudinal ridge. According to Taquet (1976, p. 131), this metacarpal is unlike that of the holotype, but this is an error; Taquet (1976) likely was referring to metacarpal V instead of III (it is a misspelling of 5 for 3) because metacarpal III of the holotype is quite similar, according to figs. 56 and 57a.

Metacarpal IV is also expanded at both extremities and has a straight shaft. The outline of its proximal surface is elliptical (the bone is craniopalmarily compressed) and is convex, while it is described as strongly convex and with a triangular outline by Taquet (1976), who is probably referring to metacarpal IV of the holotype. In fact, the proximal portion of the latter differs from that of the paratype in being mediolaterally flattened. There are no facets for the reciprocal articulation in metacarpals III and IV.

Metacarpal V is stout and hourglass-shaped in craniopalmar view. Its distal extremity is wider than the proximal one. Its proximal surface has a rounded outline and is shallowly concave in the middle. Its palmar surface is slightly concave, whereas the cranial surface is convex. The distal end is expanded palmarily, whereas the proximal end is expanded cranially. A small knob occurs near the distal extremity of the metacarpal at its palmomedial corner. The distal surface has a sub-rectangular outline, with a shallowly concave central part (it is convex according to Taquet, 1976). There are no articular facets for metacarpal IV. As observed by Taquet (1976, p. 131), metacarpal V of the holotype is completely different, being much slenderer, nearly as long as metacarpal IV and mediolaterally flattened.

Metacarpals II–IV would be separated by spatii interossei if appressed. This and the absence of articular facets between metacarpals III and IV suggests that metacarpals II–IV did not form a compact block in O. nigeriensis but were instead somewhat spreading (Fig. 15E), unlike the metacarpals of I. bernissartensis and M. atherfieldensis. Taquet (1976, fig. 53) reconstructs the palm this way too. Metacarpals of O. nigeriensis are comparatively slenderer than those of I. bernissartensis and comparatively shorter than those of M. atherfieldensis.

Manus phalanges (Fig. 15). The right manus of MSNVE 3714 preserves seven original phalanges, whereas the left one has six original phalanges. Only four phalanges were preserved in the holotype. Seven phalanges from the paratype left hand were used to complete the description of the manus according to Taquet (1976, p. 132), but those phalanges are referred to the right hand in the caption of fig. 57. The Venice specimen preserves phalanges II-3 (ungual, left), III-3 (ungual, left), IV-2 (right) and IV-3 (ungual, both right and left), which were ignored by Taquet (1976). In MSNVE 3714, both ungual phalanges of digit I are plaster copies of the only ungual phalanx of digit I preserved in the paratype (Taquet, 1976, fig. 57c; it is wrongly referred as the ungual of digit V in the caption). This copy is nevertheless worth describing because it is an important element. It is 60 mm-long, conical with a circular proximal surface. One side is flat and without grooves; it is probably the palmar side, but the phalanx is mounted in two different ways in the two hands, so its orientation is unreliable. The opposite side is deeply convex, so that the cross-section at mid-phalanx is D-shaped. The part of the phalanx facing the elbow on the left hand and the palm in the right hand has a longitudinal groove (probably a nail groove).

Phalanx II-1 (right hand) is hourglass-shaped in palmar and dorsal views, with proximal and distal extremities that are mediolaterally expanded. Its distal end is more dorsopalmarily expanded than the proximal end. Its proximal surface has a sub-circular outline and is flat; its distal surface has a sub-triangular outline and is also flat. Its dorsal surface is covered by a film of resin. This phalanx is slenderer than phalanx II-1 of the paratype figured by Taquet (1976, fig. 57b). Phalanx II-2 (right hand) is small, mediodistally short and with a slightly concave palmar surface. Its dorsal surface is also covered by a film of resin. Its proximal surface has a bean-like outline and is slightly convex; its distal surface has a sub-elliptical outline and is flat. This phalanx is proximodistally shorter and lateromedially broader (i.e., it is less blocky) than phalanx II-1 of the paratype figured by Taquet (1976, fig. 57b). Phalanx II-3 (the ungual; left hand) is elongate and hoof-like; its proximal surface has an elliptical outline and a depression at its center.

Phalanx III-1 (preserved in both hands) is a stout and hourglass-shaped element in palmar and dorsal views. Its proximal surface is flat with a sub-elliptical outline. Distally, it bears two rounded condyles that are separated by a broad and shallow intercondylar groove. The right phalanx III-1 of the paratype figured in Taquet (1976, fig. 57b) is slightly different from both phalanges III-1 of MSNVE 3417. The ungual phalanx of digit III (phalanx III-3; left hand) is hoof-like and elongate (it is the longest ungual phalanx). It shows two longitudinal nail grooves starting in the upper third of the phalanx and reaching its distal tip. Its proximal surface has a sub-elliptical outline and a concave center.

Phalanx IV-1 (right hand) is hourglass-shaped in palmar and dorsal views and its distal extremity is more mediolaterally expanded than the proximal one. Its proximal surface has a sub-elliptical outline and is flat. Its sub-triangular distal condyles are separated by a shallow intercondylar groove that is narrower than that of phalanx III-1. This phalanx is slenderer than phalanx IV-1 of the paratype figured by Taquet (1976, fig. 57b). Phalanx IV-2 (left hand) is hourglass-shaped and relatively stout in palmar and dorsal views. It is partly reconstructed and/or covered by resin. The ungual of digit IV (phalanx IV-3, preserved in both hands) is elongate and hoof-like. The proximal articular surface has an elliptical outline and a concave center.

Phalanx V-1 (preserved in both hands) is the most robust phalanx of the manus. It is hourglass-shaped in palmar and dorsal views. Its proximal extremity is more dorsopalmarily expanded than the distal one. Its proximal surface has a bean-like outline and a shallowly concave center. Its palmar surface is slightly concave and presents some thin longitudinal grooves and wrinkles (which are more evident in the right element than in the left one), possibly for muscular insertion or evidence of a cartilage cover (like the rough surface of the lateral side of the caudal vertebrae near their articular faces). Its distal surface has also a bean-like outline and is slightly convex. The phalanx V-1 figured in Taquet (1976, fig. 57b) is the phalanx mounted as left phalanx V-1 in MSNVE 3714. Phalanx V-2 (right hand) is also hourglass-shaped in palmar and dorsal views. Its proximal and distal surfaces have sub-elliptical and bean-like outlines, respectively; both have a slightly concave center.

Ilium (Figs. 16A–16E). Both ilia are preserved, but the left ilium is more complete than the right one. The left ilium is 91% the length of the ilium of the holotype. The dorsal margin of the iliac blade between the preacetabular and postacetabular processes is straight in lateral and medial views, whereas it is convex in the holotype. As anticipated in the discussion on the diagnosis, the preacetabular process is comparatively slightly shorter than that of the ilium of the holotype, preacetabular process length/total length of the ilium ratio being 0.47 and 0.50, respectively. The preacetabular process is more arched downward that that of the paratype and its rostral end is more twisted dorsally, resembling that of the specimen of I. bernissartensis figured in Norman (1980, fig. 63). The ilia were detached from the sacrum (Fig. 2, see below), thus the pelvis and sacrum were probably not fused. The different shape of the pubic peduncles in the holotype and MSNVE 3714 is caused by the incompleteness of the peduncle in both ilia of the latter. The dorsal and ventral notches at the beginning of the postacetabular process are shorter and deeper than in the holotype and the process is slenderer. The supracetabular process (“hint of an anti-trochanter” of Taquet, 1976, p. 60) is a massive bulge that protrudes laterally from the transversely thickened dorsal margin of the ilium and is located slightly more caudally with respect to the position it has in the holotype. In both cases, it is quite skewed caudally with respect to the acetabulum. The postacetabular process bears a distinct and petaloid brevis shelf that faces ventromedially. The shelf is flat, so there is no brevis fossa.

Figure 16 MSNVE 3714, pelvic girdle elements.

Left ilium in medial (A); lateral (B); dorsal (D); and ventral (E) views. Left pubis in lateral (F) and medial (G) views. Left ischium in medial (I) and lateral (J; upside-down) views. C, H and K are the drawings of the elements of figures A–B, F–G and I–J, respectively, with the reconstructed parts evidenced in dark gray colour. Abbreviations: ac, acetabulum; bs, brevis shelf; ilp, iliac peduncle of ischium and pubis; isf, distal ‘foot’ of the ischium; isp, ischial peduncle of ilium and pubis; ms, medial shelf; no, notch; obn, obturator notch; obp, obturator process; pop, postacetabular process; por, posterior pubic ramus (pubis s.s.); ppb, prepubic process; ppn, neck of the prepubic process; prp, preacetabular process; pup, pubic peduncle of ilium and ischium; sap, supracetabular process; sf, facet for the articulation with sacrum. Scale bar in A–B, F–G and I–J equals 10 cm.

Pubis (Figs. 16F–16H). In both pubes, the distal portions of the prepubic blade are much more expanded dorsoventrally than the prepubic blade of the holotype, but they are reconstructed, so this is artificial. The main apparent difference between the pubes of MSNVE 3714 and those of the holotype is the shape of the obturator foramen and related processes. The ischial peduncle of the holotype is slender and has a sub-spherical distal end. A blade-like process starts ventromedially from the basal part of the ischial peduncle and the base of the posterior pubic ramus and runs parallel to the ischial peduncle, ending at the level of its distal extremity. This process and the ischial peduncle nearly surround the obturator opening in medial view, while they form an obturator gutter in lateral view (Taquet, 1976, fig. 58b). In both pubes of MSNVE 3417, the blade-like process was broken and is missing and only the ischial peduncle remains. Consequently, the obturator ‘foramen’ appears as an open notch bordered dorsally by the ischial peduncle and ventrally by the posterior pubic ramus. The obturator opening is bordered by the ischial peduncle and by a process arising from the shaft of the posterior pubic ramus in non-hadrosaurid taxa (e.g., Lanzhousaurus magnidens (see You, Ji & Li, 2005, fig. 3); M. atherfieldensis (see Norman, 1986, fig. 55A–B); E. caroljonesa (see McDonald et al., 2012a, fig. 31)), while it is totally open in hadrosaurids (Horner, Weishampel & Forster, 2004). The obturator opening is closed (i.e., a true foramen) in Tenontosaurus tilletti (see Forster, 1990, fig. 16), Dryosaurus altus (see Galton, 1981, fig. 10) and Dysalotosaurus lettowvorbecki (see Galton, 1981, fig. 12L). It can be either open or closed in Camptosaurus dispar (see Gilmore, 1909; Verdú et al., 2017). Thus, the condition in O. nigeriensis is somewhat primitive, although the formation of the obturator foramen is obtained in a different way with respect to the non-styracosternan taxa.

Ischium (Figs. 16I–16K). Both ischia are preserved, but some portions are reconstructed, in particular the pubic and obturator processes and the distal ‘boot’. They are 105% the length of the ischia of the holotype. The iliac peduncle of MSNVE 3714 is more robust and expanded distally than that of the ilium of GDF 300 (Taquet, 1976, fig. 60). The shaft is straight, whereas it is sinuous (with the distal portion curved cranially) in the holotype.

Femur (Figs. 17A–17G). The right femur is original, while the left one is a plaster replica. Taquet (1976, p. 141) based his description of the femur of Ouranosaurus on the left femur of the paratype because those of the holotype are partially and poorly preserved. The left femur of MSNVE 3714 is just a copy of the left femur of the paratype figured by Taquet (1976, fig. 62 and pl. 24, figs. 1 and 3; it is erroneously referred to the holotype in the caption of fig. 62). The right femur is 920 mm-long, 8% longer than the left one and much longer than the corresponding tibia (129.5%). The shaft of the femur is straight in lateral and medial views except for the distal third, which gently curves caudally. In the right femur, the lesser trochanter is placed slightly more distally than in the left one. The fourth trochanter was described by Taquet (1976, p. 143) as “pendant” in Ouranosaurus, but it is not actually pendant as described in Tenontosaurus tilletti (see Forster, 1990, fig. 19b) and Dysalotosaurus lettowvorbecki (see Galton, 1981, fig. 14). The fourth trochanter of the right femur of MSNVE 3714 differs from that of the left femur (see Taquet, 1976, fig. 62c) mainly in the outline of its caudal margin, which is slightly sigmoid (Fig. 17B) in the right femur, while it is straight in the left femur. The fourth trochanter of the right femur resembles that of Hypselosaurus cf. fittoni (see Norman, 2015, fig. 18) and the trapezoid fourth trochanter reported in a ‘sub-adult’ individual of I. bernissartensis (Verdú et al., 2017). The shape of the fourth trochanter is intraspecifically variable in a sample of femora of I. bernissartensis (Verdú et al., 2017). The distal articular condyles project cranially beyond the shaft and curve medially and laterally, respectively, in a way to nearly encircle the cranial intercondylar groove (compare Fig. 17G and Taquet, 1976, fig. 62e). According to Taquet (1976, p. 143), this groove is completely encircled in the isolated femur GDF 302.

Figure 17 MSNVE 3714, hind limb: femur, tibia, fibula and tarsals.

Right femur in medial (A), lateral (C), cranial (D), caudal (E), proximal (F) and distal (G) views; B is a particular of the fourth trochanter. The left tibia and fibula in medial (H), lateral (I), cranial (J), caudal (K), proximal (L) and distal (M) views. L is the mirrored proximal view of the right tibia-fibula because the proximal part of the left tibia-fibula is poorly preserved and badly mounted. Astragalus in cranial (N) and caudal (O) views. Abbreviations: as, astragalus; ca, calcaneum; caap, caudal ascending process of the astragalus; caig, caudal intercondylar groove; cf, caput femoris (femoral head); cmc, proximal caudomedial condyle of tibia; cnc, cnemial crest; cod, condylid; crap, cranial ascending process of the astragalus; crig, cranial intercondylar groove; ftr, fourth trochanter; gtr, greater trochanter; hn, neck of the femoral head; lc, proximal lateral condyle of tibia; lcd, distal lateral condyle of femur; lm, lateral malleolus; ltr, lesser trochanter; mcd, distal medial condyle of femur; mm, medial malleolus. Scale bar equals 10 cm in A–M.

Tibia and fibula (Figs. 17H–17M). Both tibiae and fibulae are preserved; the left tibia is 90% the length of the holotype tibia. Fibulae are glued to the relative tibiae and cannot be removed. The left fibula is glued in a slightly incorrect position: it should rest in a shallow lateral depression of the proximal part of the tibia, bordered by the cnemial crest (Norman, 1986, fig. 58F; see also Godefroit et al., 1998, p. 44), not on the tibial condyles.

Pes (general). The left pes (Fig. 18A) contains original elements, whereas the right one is totally reconstructed. All pedal elements are glued together in the mounted skeleton.

Tarsals (Figs. 17H–17O). The astragalus and calcaneum are glued to the tibia and fibula. The astragalus is quadrangular, broader medially than laterally in distal view and cup-shaped in medial view. Two ascending processes occur along its proximal margin. The cranial ascending process is located in the craniolateral corner; it is broad but low (however, its apical part is damaged) and thin (Fig. 17N). The caudal ascending process occurs in the caudomedial corner; it is pointed, small and thick (Fig. 17O). In the holotype, the larger process is the caudal process and both are more centrally placed along the caudal and cranial margins of the astragalus, respectively (Taquet, 1976, p. 148, fig. 65b).

Figure 18 MSNVE 3714, hind limb: pes.

Left pes in dorsocranial view (A) and drawing with the totally reconstructed parts evidenced in dark gray colour and minor reconstructed portions in pale gray (B). Ungual phalanx of digit III in dorsal (C) and medial (D) views. Abbreviations: mtII–IV, metatarsals II–IV; ng, nail groove; pphII–IV, phalanges of pedal digits II–IV (the last phalanx is always the ungual one). Elements without abbreviation are reconstructed. Scale bar in A equals 10 cm.

Metatarsals (Fig. 18A). The metatarsus is similar to the metatarsus of the right pes of the holotype described by Taquet (1976), but it is proportionally shorter. In fact, metatarsals II–IV are 88, 73 and 83%, respectively, the length of those of the holotype. Unlike the metacarpals, the metatarsals are locked to each other to form a compact metatarsus.

Phalanges (Fig. 18). Digits II–IV have three, four and five phalanges, respectively, but phalanges II-3 and IV-2 to 4 are reconstructed. According to Taquet (1976, p. 153), seven pedal phalanges are preserved in the paratype; eight are actually present in the Venice specimen, including unguals III and IV, which should not be present according to Taquet (1976). Lengths reported by Taquet (1976) correspond with those measured in MSNVE 3714 for phalanges II-1, II-2, III-1 and III-2. All those phalanges are considered to be from the right pes in the caption of fig. 71; in the text (Taquet, 1976, p. 153), phalanges II-1 and III-1 are reported as left, III-2 as possibly right, while the provenance of phalanges II-2, IV-2 and III-3 is not established. Furthermore, phalanx IV-2 is referred to the holotype in the text, while it is assigned to the paratype in the caption of the figure. The only ungual phalanx is reported as the right phalanx IV-5 in the figure, while the text says only that it is from the right foot; that phalanx does not correspond with the ungual of digit IV of MSNVE 3714 and resembles more the ungual placed at the end of digit III.

Because of this confused original description, all of the phalanges are re-described as they are mounted in MSNVE 3714.

Phalanx II-1 is stout, gently arched laterally and hourglass-shaped in dorsoplantar view. Its distal end is more expanded lateromedially than the proximal end. Its proximal surface has a quadrangular outline and is flat; its distal end is divided into two condyles that are separated by a wide and shallow intercondylar groove. Phalanx II-2 is much smaller than phalanx II-1 and has a quadrangular outline in dorsoplantar view. Its proximal surface has a sub-triangular outline; the distal end bears two scarcely developed articular condyles.

Phalanx III-1 is large, stout and as long as it is wide. Its proximal surface has a sub-elliptical outline and is slightly concave. Its distal end bears two scarcely developed articular condyles. Phalanx III-2 is disc-like, and much proximodistally shorter than lateromedially wide. Both proximal and distal surfaces have a sub-triangular outline; the proximal one is slightly convex, whereas the distal one is concave. Phalanx III-3 has a shape similar to that of phalanx III-2, but it is transversely narrower. The long phalanx III-4 (the ungual phalanx, 70 mm long) is spade-like, dorsoplantarily flattened and slightly arched (Figs. 18C–18D). The medial expansion of its distal end is more developed than the lateral one, but it has a rough aspect and could be broken. There is a very shallow nail groove along the medial expansion.

Phalanx IV-1 is hourglass-shaped in dorsoplantar view and resembles phalanx II-1. Its proximal end is more dorsoplantarily expanded than the distal end. Its proximal surface has a sub-rectangular outline. The distal end bears two condyles that are separated by a wide and shallow intercondylar groove. The ungual phalanx IV-5 is gently arched plantarily and has a squared outline in dorsoplantar view, but its distal portion is missing and it was possibly like ungual phalanx III.

Phylogenetic Analysis

There is no agreement on the phylogenetic relationships of Ouranosaurus nigeriensis (e.g., Sereno, 1986; Norman, 2004; Norman, 2015; McDonald, Barrett & Chapman, 2010; McDonald, Wolfe & Kirkl, 2010; McDonald et al., 2012b). In his phylogeny of the Ornithischia, Sereno (1986) found Ouranosaurus to be a member of the Hadrosauroidea and sister taxon of the Hadrosauridae. According to Norman (2004), O. nigeriensis is more derived than Iguanodon bernissartensis and Mantellisaurus atherfieldensis, and basal to non-hadrosaurid iguanodontians (Probactrosaurus gobiensis, Eolambia caroljonesa, Protohadros byrdi and Altirhinus kurzanovi). In McDonald et al.’s analysis (2012b, fig. 10), O. nigeriensis is more derived than I. bernissartensis and basal within the Hadrosauroidea. In the latest published phylogeny by Norman (2015, fig. 48), it occurs in a more basal position within the Styracosterna as the sister taxon of the clade ‘Iguanodontoids’ + Hadrosauriformes.

Using this new information from the Venice specimen, we performed a cladistics analysis to reassess the phylogenetic position of Ouranosaurus nigeriensis. Obviously, only characters regarding the postcranium could be changed because MSNVE 3714 lacks the skull and the lower jaw.

We used the data matrix of McDonald (2012; “second run” on p. 2) with the addition of scores from McDonald et al. (2012b) for Proa valdearinnoensis. Character 77 (78 in the Nexus matrix because the character numbering starts from 1, while the matrix by McDonald (2012) starts from 0 because it was created with TNT) is missing in the codings for Proa valdearinnoensis given by McDonald et al. (2012b); see tab. 1), supposedly because of a typo, thus, we consider it as unknown (coding ?). Forty out of 135 (30%) characters could be potentially changed based on the new information from the Venice specimen. We changed the codings of characters 97 (which becomes 98 in the Nexus matrix); 100 (101); 106 (107); 111 (112); 119 (120); 122 (123); and 132 (133) (see Supplemental Information 4). Characters 98 (99) and 101 (102) were deleted (see Supplemental Information 4). The resulting matrix has 133 characters and 62 taxa. The analysis was run using the default options of PAUP ∗ v. 4.0b10 (Swofford, 2002) with maxtrees reset to 50,000. It yielded 50,000 equally most parsimonious trees (tree length: 387; consistency index: 0.5013; retention index: 0.8169; rescaled consistency index: 0.4095); the strict consensus tree is poorly resolved and gives no information on the phylogenetic position of O. nigeriensis The Adams consensus tree (Fig. 19) is more resolved and Ouranosaurus occupies the same position as in the tree published by McDonald et al. (2012b, fig. 10). However, the tree differs from that by McDonald et al. (2012b) in the position of Elrhazosaurus nigeriensis and Valdosaurus canaliculatus, Uteodon aphanoecetes and Cumnoria prestwichii, Planicoxa venenica, Proa valdearinnoensis and Jinzhousaurus yangi (Fig. 19). The position of O. nigeriensis is the same also in the agreement subtree obtained with TNT (Goloboff, Farris & Nixon, 2008) (search protocols: 1,000 addition sequence replications of the “Traditional Search” analysis, using Tree Bisection Reconnection—TBR—as swapping algorithm, and saving 10 trees per replication; the recovered tree islands was explored performing a second round of TBR, saving all the shortest trees found; memory limit (maxtrees) was set at 50.000; statistics for the trees are the same obtained with PAUP).

Figure 19 Phylogenetic relationships of Ouranosaurus nigeriensis.

Adam consensus tree obtained from the revised matrix of McDonald et al. (2012b). Legend: 1, Iguanodontia; 2, Dryomorpha; 3, Ankylopollexia; 4, Styracosterna; 5, Hadrosauroidea.

Using the data matrix from Norman (2015), characters that could be potentially changed based on the Venice specimens are 36 out of 105 (34%). Characters 80 and 85 are redundant. Character 85 was kept after emendation, while character 80 was replaced with a new character; we slightly modified the character states in characters 71, 92 and 94 (see Supplemental Information 4). The codings of characters 73, 81, 83, 87, 90, 92, 95, 99 and 105 were changed. The modified matrix was analysed using PAUP ∗v. 4.0b10 (Swofford, 2002) using the default options. All characters were given equal weighting and run unordered. The analysis yielded three equally most parsimonious trees (tree length 315; consistency index: 0.5683; retention index: 0.7733; rescaled consistency index: 0.4394). The strict consensus tree has the same topology as the strict consensus tree of Norman (2015). The differences between the two analyses are the tree length (315 and 313, respectively) and the addiction of some state changes between the node 48 ->Ouranosaurus nigeriensis (90 1->3; 92 0->2; 99 0->1; and 105 1->2 in the emended analysis (see Supplemental Information 4).

Revision of the phylogenetic analyses by McDonald et al. (2012b) and Norman (2015) is beyond the scope of this paper. The basal position of O. nigeriensis within the Styracosterna found by Norman (2015) seems to be supported by some character states (most of which were not considered in the two matrices) occurring basally within the iguanodontians: 15 dorsal vertebrae, small size of the manus, spreading palm, digit I that is not perpendicular to the axis of the manus, nearly closed obturator opening of the pubis and size of the ascending processes of the astragalus (although in the holotype only). The ‘hadrosaurid’ features (elongate skull with laterally expanded and dorsoventrally flattened terminal part of the rostrum, oral margin of the premaxilla reflected dorsally to form a distinct rim and long ‘diastema’ in the dentary) were probably convergently acquired by O. nigeriensis.

Figure 20 MSNVE 3714, left humerus, thin section.

Panoramic view under lambda filter (the outer surface of the bone is at the top of the figure) (A); detail of the progressive transition between the compacta and the medullary cavity, characterized by erosional cavities (B); detail of the fibrolamellar bone and longitudinal vascularization forming the primary bone (C); detail of the microstructure of the outermost cortex showing the absence of an EFS (D); no secondary osteons are observed in the inner cortex. Green arrows point to the LAGs. Abbreviations: lb, lamellar bone; mc, medullary cavity; ps, periosteal surface; vc, vascular canals; wb, woven bone. Scale bars equal 10 mm in A and 1 mm in B–D.

Osteohistology of MSNVE 3714

Humerus

The left humerus was sampled on its caudal side at mid-shaft. The cylindrical core sample was 23 mm thick and 14 mm in diameter (Fig. 20A). The medullary cavity shows relatively few trabeculae, which are generally not connected to each other. Trabecular density is lower in comparison to that present in the femur (see below). The edge of the medullary cavity is neat (Fig. 20B). Erosional cavities are present in the inner compacta; they show a lower density than in the femur (see below). Those cavities have an elliptical or rounded outline and occur in the inner one fifth of the compacta; their density decreases moving toward the outer surface of the cortex. The microstructure of the cortex is fibrolamellar with a matrix composed of woven bone (Figs. 20C–20D). Compacted coarse cancellous bone (CCCB; sensu Hübner, 2012) occurs in the inner cortex surrounding the erosional cavities (Fig. 20B). Vascularization (sensu lato; see Chinsamy, 2005) is mainly composed of well-developed (i.e., large and with many lamellae) primary osteons that become more and more organized (i.e., regularly arranged in the space) towards the outer cortex (Figs. 20C–20D). Vascularization has a laminar circumferential arrangement (Figs. 20C–20D). The distance between the single vascular canals is relatively high (Figs. 20C–20D). Zonation is present. Four to six LAGs are recognized in the compacta (Fig. 20A); the spacing between successive LAGs decreases moving towards the outer surface of the bone. Secondary osteons and an EFS are absent (Figs. 20A–20D).

Femur

The right femur was sampled on the craniolateral side at mid-diaphysis. The cylindrical core sample was 21 mm thick and 14 mm in diameter (Fig. 21A). The medullary cavity is characterized by isolated trabeculae. The transition between the medullary cavity and the compact cortex is gradual because of the presence of many resorption cavities in the inner cortex (Fig. 21B). Resorption cavities tend to decrease in density moving towards the external surface of the cortex, and their outline changes from irregular to rounded or elliptical. The compacta is composed of two different types of bone: the primary bone and the CCCB. The primary bone is composed of fibrolamellar bone with woven bone forming the matrix (Figs. 21C–21D). CCCB bone is present in the inner most compact bone wall, especially in the areas surrounding the resorption cavities (Fig. 21B). Primary osteons are abundant throughout all the compact bone.

Vascularization is irregularly organized and a clear orientation is not evident. Primary vascular canals are still open, although infilling of lamellar bone is present. Secondary osteons (Haversian systems) cannot be identified in the thin section. No LAGs or annuli can be observed and there is no EFS (Figs. 21A–21D).

Figure 21 MSNVE 3714, right femur, thin section.

Panoramic view under lambda filter (the outer surface of the bone is at the top of the figure) (A); the gradual transition between the compacta and the medullary cavity made of CCB and erosional cavities (B); detail of the plexiform vascularization and fibrolamellar bone forming the primary bone and the absence of zonation and LAGs within the compacta (C); detail of the microstructure of the outermost cortex showing the absence of an EFS (D). Abbreviations: ec, erosional cavities; lb, lamellar bone; ps, periosteal surface; vc, vascular canals; wb, woven bone. Scale bars equal 10 mm in A and 1 mm in B–D.

Tibia

The right tibia was sampled craniolaterally in the diaphysis, slightly below mid-shaft. The cylindrical core sample was 26 mm thick and 14 mm in diameter (Fig. 22A). Within the medullary cavity, a typical spongiosa is absent: trabeculae are low in density (Figs. 22A–22B). The boundary between the medullary cavity and the cortex is abrupt and uneven and there is a thin endosteal lamella. The primary bone microstructure is fibrolamellar with woven bone constituting the matrix (Figs. 22C–22D). In the inner cortex, vascularization is irregular in its orientation, density and organization. Primary osteons are well developed and generally have a laminar circumferential orientation. Locally, reticular arrangement of the primary vascular canals is observed. Organization of primary vascular canals increases towards the outer surface of the bone. Infilling of lamellar bone is present in those canals, which, however, are not completely filled. Secondary osteons are abundant in the innermost cortex, extending over one fifth to one sixth of the compact bone wall thickness (Fig. 22A). Unlike all other sampled long bones, erosional cavities are absent. Six LAGs occur in the compact bone wall. There is no EFS (Fig. 22D).

Figure 22 MSNVE 3714, right tibia, thin section.

Panoramic view under lambda filter (the outer surface of the bone is at the top of the figure) (A); gradual transition between the compacta and the medullary cavity made of CCB and erosional cavities (B); detail of the deeper cortex, showing zonation of the primary bone, irregular vascularization and fibrolamellar bone (C); detail of the outermost cortex showing the absence of an EFS (D); note the remodeling in the inner compacta. Abbreviations: lb, lamellar bone; mc, medullary cavity; ps, periosteal surface; so, secondary osteons; vc, vascular canals; wb, woven bone. Green arrows point to the LAGS. Scale bars equal 10 mm in A and 1 mm in B–D.

Neural spine

The neural spine of dorsal vertebra 14 was cross-sectioned within the basal third, in the middle, and within the apical third (Figs. 23A, 24A and 25A). The cross-section is oval proximally becoming more rectangular in the central and distal segments. The medullary cavity is filled with spongiosa (Figs. 23B, 24B and 25B). The boundary between the medullary cavity and the cortex is gradual, as erosional cavities occur between the trabecular structure of the medullar cavity and the compact cortex (Figs. 23C, 24C and 25C). The compacta becomes progressively thinner moving from the proximal part of the spine towards the apical part. The microstructure is fibrolamellar and tends to become more organized moving towards the outer surface of the bone. Evidence of the presence of Sharpey’s fibers is observed on the lateral surfaces of the spine in the proximal section. No Sharpey’s fibers are found in the other two thin sections (middle and apical). Primary vascularization is mostly longitudinal and becomes more organized and low in density moving through the outer cortex towards the outer surface. Haversian systems are generally present in the inner half of the compacta (Figs. 23B–23C, 24B–24C and 25B–25C). Six, four, and three LAGs are identified in the proximal, median and distal sections, respectively. We consider six or seven LAGs the most reliable count to establish the age of the individual because the base of the neural spine is expected to preserve the most complete growth record. The spacing between the zones decreases towards the outer surface of the bone. An EFS is absent (Figs. 23D, 24D and 25D).

Figure 23 MSNVE 3714, base of the neural spine of dorsal vertebra 14, transverse thin section.

Panoramic view of the cranial half of the section (A); Haversian systems in the innermost cortex (B); transition between the compacta and the medullary cortex with erosional cavities and remodeling (C); detail of the outer cortex showing absence of an EFS and outermost LAGs (D). Abbreviations: lb, lamellar bone; ps, periosteal surface; so, secondary osteons; tb, trabeculae; vc, vascular canals; wb, woven bone. Green arrows point to the LAGS. Scale bars equal 10 mm in A, 1 mm in B and C, and 500 µm in D.

Figure 24 MSNVE 3714, middle neural spine of dorsal vertebra 14, transverse thin section.

Panoramic view of the cranial half of the section (A); Haversian systems in the inner most cortex and endosteal bone (B); gradual transition between the compacta and the medullary cortex with erosional cavities and marked remodeling of the primary bone (C); detail of the outer cortex showing absence of an EFS and zonation of the primary bone (D). Abbreviations: eb, endosteal bone; ec, erosional cavities; ps, periosteal surface; so, secondary osteons; tb, trabeculae; vc, vascular canals; wb, woven bone. Green arrows point to the LAGS. Scale bars equal 10 mm in A and 1 mm in B–D.

Figure 25 MSNVE 3714, apex neural spine of dorsal vertebra 14, transverse thin section.

Panoramic view of the cranial half of the section; note how the compact cortex becomes thinner trending through the top of the neural spine (A); detail of the Haversian systems in the inner most cortex and endosteal bone (B); gradual transition between the compacta and the medullary cortex (C; note the erosional cavities and deep remodeling of the primary bone); detail of the outer cortex with outermost LAGs but without an EFS (D). Abbreviations: eb, endosteal bone; ec, erosional cavities; lb, lamellar bone; ps, periosteal surface; so, secondary osteons; wb, woven bone. Green arrows point to the LAGS. Scale bars equal 10 mm in A and 1 mm in B–D.

Dorsal rib

The transverse cross-section of the proximal part of dorsal rib 15 has an oval outline. The cortex of the lateral side is heavily eroded; its maximum thickness is 17 mm (Fig. 26A). The medullary cavity is filled with spongiosa. The boundary between the spongiosa and the compacta is gradual, with the erosional cavities in the inner cortex becoming smaller and fewer moving toward the outer surface of the cortex (Fig. 26B). The microstructure is fibrolamellar with a matrix of woven bone. Primary osteons are well developed and abundant in the outer cortex and have a longitudinal orientation. Vascular canals are partially infilled with lamellar bone, so they are still open. Primary vascularization is more organized and less dense in the outer cortex. Secondary osteons are present in the innermost cortex (Fig. 26C). Six or seven LAGs can be identified (Fig. 26A); they tend to become increasingly closely spaced moving towards the outer surface of the cortex. An EFS is absent (Fig. 26D).

Figure 26 MSNVE 3714, dorsal rib, transverse thin section.

Panoramic view of the thin section of the dorsal rib (A); transition between the outer cortex and the medullary cavity (B; note erosional cavities and deep remodeling of the primary bone); Haversian systems in the inner cortex (C); detail of the outermost cortex showing zonation and the absence of an EFS (D). Abbreviations: ec, erosional cavities; ps, periosteal surface; so, secondary osteons; vc, vascular canals; wb, woven bone. Green arrows point to the LAGS. Scale bars equal 10 mm in A, 1 mm in B and D, and 500 µm in C.

Discussion

Is MSNVE 3714 a composite?

The Venice specimen undoubtedly includes the paratype of O. nigeriensis indicated in Taquet (1976, p. 58), figured in Taquet (1976, pl. 9, fig. 2) and mapped in one of the two sheets (sheet 1) that we received from the MNHN, where it is labeled “Ouranosaurus nig[eriensis]—Airfield—1970—(specimen Venice Museum pro parte)”. However, the words “pro-parte” could mean that (1) only part of the mapped bones was used in the mount of the Venice skeleton, while the other elements remained at the MNHN or were sent back to the Musée National du Niger (now Musée National Boubou-Hama) with the holotype; (2) the paratype material was completed with bones from another individual to mount a more complete skeleton; (3) both 1 and 2 are true. Sheet 1 was associated with a second sheet, containing the map of a different individual (Fig. 2), whose relationship with the paratype is unclear. Possibly, some elements from that individual were collected and used to complete the Venice mount.

Some incongruity exists between MSNVE 3714 and the mapped paratype skeleton. As anticipated above, a maximum of 15 dorsal vertebrae are identifiable in the map of the paratype (Fig. 2), thus MSNVE 3714 has at least two dorsal vertebrae in excess. It is unclear whether those two vertebrae came from the specimen mapped in the sheet 2 or from another source. They were added to the paratype material to reach the number of 17 dorsals because Ouranosaurus was supposed to have 17 dorsals based on the dorsal count in Mantellisaurus atherfieldensis (17) and the supposed count in Iguanodon bernissartensis (18; actually, there are 16 plus a dorsosacral, according to Norman, 1980) (Taquet, 1976, p. 111 and 121).

In the map, only three neural spines can be seen in the region of the sacrum (the drawing of a distal fourth was marked as it were drawn in error; see Fig. 2), while the sacrum of MSNVE 3714 is composed of six fused vertebrae. However, it is plausible that the other three sacral vertebrae were not visible because they were still embedded in rock and were not mapped.

The number of caudal vertebrae (33) correspond to the map of the paratype and to MSNVE 3714, only when assuming that the element identified as a phalanx in the handwritten note on the map was a further centrum (which it probably was because of size and position of the bone and its alignment with the vertebral segment) and also considering in the count the vertebra that was not mapped (see Fig. 2). The five vertebrae that are totally reconstructed in MSNVE 3714 would correspond with those missing in the gap between caudal 24 and the following caudals (see Fig. 2). However, the reconstructed elements are the caudal vertebrae 27–31 in MSNVE 3714 (see Fig. 4D). So, some elements could be misplaced.

Only the pedicels of the first two chevrons are drawn in the field map of the paratype, between caudals 3 and 4 and 4 and 5, respectively (Fig. 2). The first two chevrons are complete in MSNVE 3714. This would suggest that those chevrons were replaced with material from another individual, possibly from that figured on sheet 2 (Fig. 2).

Only one element is identified as a coracoid by the handwritten notes in the field map of the paratype. It is not specified whether it a left or a right one, but the paratype preserved the right one according to Taquet (1976). MSNVE 3714 has also the left coracoid, while the right one is a copy of the paratype’s right coracoid (see below). Possibly, a broad element that is close to the coracoid in the map and is labeled as “sternum?” (Fig. 2) could be the other coracoid. Alternatively, the left coracoid could be missing in the map because it was still embedded in the rock, but this remains speculative in absence of a list of the bones obtained after preparation of the collected blocks in the MNHN laboratories (a list was never compiled; P Taquet, pers. comm., 2017). There is no coracoid in the map on sheet 2. The smaller size of the left coracoid with respect to the right one suggests that it is not the coracoid GDF 301 of the referred material, which is reported by Taquet (1976 , p. 58) as “coracoïde de grandes dimensions” (large-sized coracoid).

Both radius-ulna pairs are preserved in MSNVE 3714. The map of the paratype shows one radius-ulna pair and a splint of bone that is identified as a second ulna (Fig. 2). Possibly, the rest of the second ulna and the relative radius were still embedded in rock when mapped, but this remains speculative in absence of further information. The map on sheet 2 does not contain any apparent radius or ulna.

Only two rounded elements are labeled as “carpals” in the field map of the paratype, while the Venice specimen has three presumed carpals. However, it is possible that the smaller element of the carpus is the fragment of bone (bone n. 104; see Supplemental Information 3) that is adjacent to the other two.

Eleven bones at maximum can be referred to the manus in the map. They include three or four metacarpals and two or three ungual phalanges, one phalanx occurring 2.5 m away from the others (Fig. 2). MSNVE 3714 has 17 original metacarpals and phalanges, including all right metacarpals and four ungual phalanges. Thus, the number of manus elements in the map and in the mount do not correspond. Some elements could be lacking in the map because they were totally embedded in the rock or covered by other elements, but this remains speculative in absence of a list of the bones obtained after preparation of the collected blocks in the MNHN laboratories. Three right phalanges referred to the paratype in Taquet (1976) are unlike those present in MSNVE 3714. They were possibly replaced with other phalanges in MSNVE 3714, but the slight differences between the phalanges figured in Taquet (1976) and those mounted in MSNVE 3714 could also be ascribed to the inaccuracy of the drawings. No elements from the manus can be recognized in the map on sheet 2.

Only one skeletal element can be an ilium in the paratype field map (Fig. 2); it was labeled as “ilium” in a handwritten note that, however, was marked as if it were an error. Notwithstanding, that element can plausibly be an ilium because of its outline and position. It seems to lack the long preacetabular process like the right ilium of MSNVE 3714. Possibly, the left ilium is missing in the map because it was still completely embedded in the rock at the time the map was drawn, but this remains speculative, as already noted above. A complete ilium occurs in the map on sheet 2 (Fig. 2) and preserves the long preacetabular process as the left ilium of MSNVE 3714 (Figs. 16A–16E). It could have been used to complete the paratype skeleton.

MSNVE 3714 preserves both pubes, although they are incomplete distally. Only one element is labeled as “pubis” in the paratype field map; it preserves most of the long and thin posterior pubic ramus (Fig. 2), thus it can be the right pubis of MSNVE 3714 (the process is mostly reconstructed in the left pubis; Figs. 16F–16H). Two ilia occur in the map on sheet 2; the left one could be the left pubis of MSNVE 3714.

Only one element is labeled as “ischium” in the paratype field map (Fig. 2); it is close and parallel to the left fibula and does not have the outline of an ischium. One ischium (possibly two overlapping ischia) is close to the paired pubes in the map on sheet 2. MSNVE 3714 preserves both ischia; one or even both of them could come from the individual mapped on the sheet 2.

According to the field map, only the left femur was preserved in the paratype. No femur is documented in sheet 2. MSNVE 3714 also preserves the right femur, which, therefore, comes from another source. It cannot be the referred femur GDF 302 because the cranial intercondylar groove is completely encircled in that specimen (Taquet, 1976), unlike the right femur of MSNVE 3714. The different length, different shape of the fourth trochanter and the osteohistological analysis support the hypothesis that the right femur is from an individual that is distinct from the paratype. According to Horner, De Ricqlès & Padian (1999), the best growth record of the hadrosaur Hypacrosaurus is preserved in the femur, followed by the tibia, dorsal ribs and neural spines. The humerus, tibia and dorsal rib growth records in MSNVE 3714 are comparable to each other suggesting that they have a same growth stage and could belong to the same individual. The neural spine shows one or two LAGs more than the humerus, the tibia and the dorsal rib, but this is probably due to the fact that it is not as remodeled, with fewer Haversian systems. Thus, the neural spine could also belong to the same individual as the humerus, tibia and dorsal rib. Following Horner, De Ricqlès & Padian (1999), the femur should present a higher number of LAGs, higher density of Haversian systems and a more advanced remodeling respect to humerus, tibia, dorsal rib and dorsal neural spine, if it is at the same growth stage, but it does not. Its lack of outer LAGs and an EFS could be explained with superficial abrasion, but weathering cannot account for the total absence of LAGs. Of course, we do not think that the cortex of a 920 mm-long femur could be produced in a single year. Possibly, LAGs are absent because we sampled a part of the bone where they are missing. However, the lack of Haversian systems and the dense vascularisation still suggest that the femur belongs to a more immature individual than that from which the humerus, tibia, neural spine and dorsal rib come from.

The field map of the paratype shows only one tibia and fibula, plausibly the left one because of its position with respect to the left femur. MSNVE 3714 preserves both tibiae. A tibia and at least part of a fibula occur in the map on sheet 2 and might be the other crural elements of the Venice specimen.

The field map of the paratype shows elements only from the left pes (Fig. 2), including the astragalus, calcaneum, a large metatarsal and some phalanges. Two much smaller elements the size of the phalanges from the first row are identified as metatarsals by handwritten notes. Only the left pes lacking phalanges II-3 and IV-2 to 4 is preserved in MSNVE 3714. The unclear description of the pedal phalanges in Taquet (1976) does not help in understanding whether some phalanges of the paratype were replaced with other phalanges from another source in MSNVE 3714 or whether missing phalanges where just replaced. Both possibilities cannot be discarded.

Recapitulating, the field map of the paratype does not show the following elements that occur in MSNVE 3714: two dorsal vertebrae, the first two chevrons, the left coracoid, one ilium (probably the left one), one pubis, at least one ischium, the right femur and the right tibia and fibula. Therefore, MSNVE 3714 is most probably a composite, although it is mostly composed by the paratype of Ouranosaurus nigeriensis. The pelvic region of the latter was possibly completed with the pelvic elements from the individual mapped on sheet 2. The right tibia-fibula and the first two chevrons could also come from the specimen mapped on sheet 2, while no source can be hypothesized for the right femur. The two dorsal vertebrae in excess could come from another specimen or from the bone set mapped on sheet 2.

The left femur and the right coracoid of the paratype were replaced in MSNVE 3714 with plaster copies. One original ungual phalanx of manus digit I (the spike-like ungual of the pollex) was replaced with a plaster copy. The left femur was sent back to the MNBH with the holotype: it can be seen in a photo of the skeleton exhibited at the MNBH that is available in the web (http://www.gettyimages.it/detail/fotografie-di-cronaca/herbivorous-dinosaur-skeleton-of-ouranosaurus-fotografie-di-cronaca/543868764#herbivorous-dinosaur-skeleton-of-ouranosaurus-nigeriensis-taqueti-picture-id543868764). This may be the case for all the other replaced paratype bones, which were plausibly sent back to the MNBH with the holotype in order to complete the specimen and keep all of the described material in the same place.

Ontogenetic stage of MSNVE 3714

Mature individuals of Ouranosaurus were supposed to attain a length of 6–7 m (Norman, 2015, p. 62). However, the ontogenetic stage of the two known skeletons of O. nigeriensis was never reliably established. Taquet (1976) does not discuss the ontogenetic stage of GDF 300, but reports only that the elements of the axis and the bones of the neurocranium of the holotype are fused and their boundaries are difficult to recognize. Possibly, these features were considered as evidence of maturity. However, the other elements of the skull were not fused, no information about the obliteration of neurocentral sutures in other vertebrae is given in Taquet (1976) and no osteohistological investigation was ever attempted in the holotype. Furthermore, the universal validity of the obliteration of the cranial sutures as evidence of osteological maturity has been argued against by Bailleul et al. (2016). Possible evidence of osteological maturity of the holotype could be the fusion of some cervical ribs with the corresponding parapophysis of the vertebra and the co-ossification of the scapulae with their respective coracoids (Taquet, 1976, p. 108 and 122). However, Norman (2004), p. 428) considers the fusion of scapula and coracoid as “either an aberrant pathology or an autapomorphy”. Since this co-ossification does not occur in MSNVE 3714, it is not an autapomorphy of O. nigeriensis, unless it happened late in the life history of this taxon because MSNVE 3714 was not fully grown (see below). The only way to solve the dilemma is by undertaking the osteohistological study of the holotype; if it is shown to be a subadult, the pathological nature of the fusion would be supported, otherwise fusion as a consequence of growth could not be discarded.

The holotype was estimated to be 7 m long (Taquet, 1976, p. 175), while the length of the mounted MSNVE 3714 is about 6.5 m from the tip of the snout to the last preserved caudal vertebra. Although this length is lower than the original total length of the complete skeletons because the distal part of the tail is not preserved, the known lengths of MSNVE 3714 and the holotype are comparable because they have a similar number of preserved caudals (43 and 40, respectively, including those reconstructed). Thus, the total length of MSNVE 3714 is about 90% that of the holotype. The humerus, ulna and tibia of the Venice specimen are 92, 88 and 90% the length of those of the holotype, respectively, in agreement with the total length. MSNVE 3714 is therefore about 10% smaller in linear size than the holotype. Relative body proportions of the two individuals can be seen in Fig. 27. The lengths of humerus, ulna and tibia in a sample of Iguanodon bernissartensis skeletons, that are supposed to have the same ontogenetic (adult) stage, range 72–84 cm, 56–67 cm and 77–105 cm, respectively (Verdú et al., 2017). Thus, the smallest adult humerus, ulna and tibia are 86%, 84% and 73% the length of the largest adult humerus, ulna and tibia, respectively. By comparison, the different size of the two O. nigeriensis skeletons alone is not enough to support their belonging to two distinct ontogenetic stages.

Figure 27 Size of the holotype (GDF 300) and MSNVE 3714.

Holotype (A) and MSNVE 3714 (B). Redrawn and modified from Taquet (1976). Scale bar equals 50 cm.

The ontogenetic stage of MSNVE 3714 was established by observation of macroscopic evidence of osteological immaturity in the skeletal elements and by their osteohistological features. The neurocentral sutures in proximal and middle caudal vertebrae up to caudal 25 (Figs. 11B–11D) and in at least the last cervical vertebra (Fig. 5A) are not obliterated. The rough surface texture of the centrum near its articular facets in caudals 11–25 (Fig. 11A) is indicative of incomplete ossification. This is suggestive of immaturity (Bennett, 1993; Brochu, 1996; Irmis, 2007). The osteological immaturity of the paratype is supported also by the unfused ilia and sacrum, which is also observed in the holotype.

The increasing organization of vascular canals toward the outer surface, the presence of Haversian systems, the decreasing spacing between LAGs and the absence of an EFS observed in the thin sections of tibia, neural spine and dorsal rib suggest that these skeletal elements belonged to a sub-adult individual (sensu Horner, De Ricqlès & Padian, 2000). In the humerus, the increasing organization of vascular canals toward the outer surface, the decreasing spacing between LAGs, the absence of EFS, but also the absence of Haversian systems, is consistent with a sub-adult growth stage sensu Horner, De Ricqlès & Padian (2000).

The conclusions are that an over 6.5 metres-long individual of O. nigeriensis was not fully grown, although probably close to adult size.

In crocodiles, the obliteration of the neurocentral suture during ontogeny starts in the tail and ends in the neck (posterior-anterior sequence of neurocentral closure; Brochu, 1996). This is the case of most, but not all (e.g., Zheng et al., 2012), ornithopods. In contrast, that pattern was not followed in several saurischian dinosaurs (Irmis, 2007). There is considerable variation of both the sequence and timing of neurocentral suture closure within archosaurs (Irmis, 2007). The condition in Ouranosaurus suggests that it did not follow a simple posterior-anterior or anterior-posterior sequence of neurocentral closure; furthermore, obliteration occurred relatively late in ontogeny, when the individual was close to somatic adulthood

Differences from the holotype

Most of the differences between MSNVE 3714 and the holotype are minor differences that can be ascribed to intraspecific variability (individual or ontogenetic variation), as also reported in a statistically significant sample of Iguanodon bernissartensis individuals (Verdú et al., 2017). These include the outline of the neural spine of the axis, some minor differences in the dorsal vertebrae, the comparatively larger scapulae and sternals in MSNVE 3714, the major or minor robustness of the humerus and broadness of its deltopectoral crest, the different distal projection of the ulnar condyle of humerus respect to the radial condyle, the major or minor development of the olecranon of ulna, the position of the supracetabular process of ilium, the relative length and arching of the preacetabular process and the dorsal twisting of its distal end, the major or minor slenderness of the postacetabular process of ilium and the morphology of the dorsal and ventral notches at the beginning of this process, the major or minor robustness of the iliac peduncle of the ischium and the comparatively shorter metatarsals in MSNVE 3714. In the literature, the curvature of the dorsal margin of the ilium above the pubic and ischial peduncles and the acetabulum, the curvature of the shaft of the ischium, and the presence and size of the ascending processes of the astragalus are considered to be phylogenetically important within the Iguanodontia. However, the different conditions in the two skeletons of O. nigeriensis can only be explained with intraspecific variability and caution should be taken as to the phylogenetic value of those characters. The curvatures of the dorsal margin of the ilium and the shaft of the ischium are intraspecifically variable also in I. bernissartensis (see Verdú et al., 2017).

Other differences are probably caused by mistakes in the preparation or assemblage of the skeletal elements in both specimens. These mistakes include the morphology of the carpus (which is possibly made of fragments of other bones in MSNVE 3714) and the morphology of neural spines and chevrons.

The differences in relative height of the spines of the sacral vertebrae could also be ascribed to individual variability or even sexual dimorphism. However, the steps in the ‘sail’ outline in correspondence of the passage dorsal vertebrae-sacrum and sacrum-caudal vertebrae in the holotype are puzzling and could be an artifact.

The presence of five or six more proximal caudals in MSNVE 3714 than in the holotype (four to five more than in Iguanodon and Mantellisaurus) could also be ascribed to the individual variability; however, some proximal caudals could be missing in the holotype. Some middle caudals are missing in the holotype to account for the low number of preserved middle caudal vertebrae.

The metacarpal V is a stout element in relatively primitive iguanodontians such as Dryosaurus (see Galton, 1981, fig. 6I) as well as in I. bernissartensis (see Norman, 1980, fig. 60) and M. atherfieldensis (see Norman, 1986, figs. 51–52). Therefore, the morphology of the metacarpal V in the holotype is aberrant or the identification of the bone as metacarpal V is wrong. In fact, Taquet 1976 (fig. 53) used the metacarpal V from the paratype in the reconstruction of the manus of Ouranosaurus.

Possibly, the ischia of the Venice specimen are longer than those of the holotype because they are not from the paratype, but from a larger individual.

Osteohistological comparison with other ornithopods

The osteohistology of many ornithopod taxa has been studied, including Hypsilophodon (Reid, 1984; Chinsamy, Rich & Vickers-Rich, 1998); the “Proctor Lake ornithopod” (Winkler, 1994); Orodromeus (Horner, Padian & De Ricqlès, 2001; Horner et al., 2009); a “hypsilophodontid” from Dinosaur Cove/Flat Rocks, Victoria, Australia (Chinsamy, Rich & Vickers-Rich, 1998; Woodward et al., 2011); Rhabdodon (Nopcsa, 1933; Reid, 1984; Reid, 1985; Ösi et al., 2012); Mochlodon (Ösi et al., 2012); Zalmoxes (Benton et al., 2010; Ösi et al., 2012); Dryosaurus (Horner, Padian & De Ricqlès, 2001; Horner et al., 2009); Dysalotosaurus (Chinsamy, 1995; Hübner, 2012); Valdosaurus (Reid, 1984); Camptosaurus (Horner et al., 2009); Tenontosaurus (Werning, 2012); Iguanodon (De Ricqlès, Godefroit & Yans, 2012); ‘Telmatosaurus’ (Benton et al., 2010); Edmontosaurus (Reid, 1985); Maiasaura (Barreto et al., 1993; Barreto, 1997; Horner, De Ricqlès & Padian, 2000; Horner, Padian & De Ricqlès, 2001; Woodward et al., 2015); and Hypacrosaurus (Horner, De Ricqlès & Padian, 1999; Cooper et al., 2008).

Orodromeus shows longitudinal arrangement of the vascular canals, far less dense and complex than that present in more derived ornithopods like Maiasaura and Hypacrosaurus (Werning, 2012). Remodeling is scarce at the adult ontogenetic stage (Horner et al., 2009). LAGs occur in juvenile individuals; the highest observed number of LAGs is two in sections with EFS (Horner et al., 2009; Werning, 2012). Orodromeus is therefore characterized by slow growth (Horner et al., 2009; Werning, 2012). Rhabdodontids (Zalmoxes and Rhabdodon) and Tenontosaurus show higher growth rates than Orodromeus, as suggested by woven bone forming the microstructure, more complex orientation of the vascular canals (radial or circumferential) and the absence of LAGs during young ontogenetic stage (Werning, 2012). Remodeling is generally present only during late growth (late sub-adult and adult ontogenetic stages) (Werning, 2012). Moreover, vascularization generally tends to decrease through ontogeny, showing a progression between the initial rapid growth characterized by woven bone to the later slow growth indicated by lamellar bone leading to an EFS (Werning, 2012). More derived iguanodontians (e.g., Dryosaurus, Dysalotosaurus, Valdosaurus, Camptosaurus, Iguanodon, Telmatosaurus, Edmontosaurus, Maiasaura and Hypacrosaurus) show the presence of rapidly depositing woven bone and vascular canals with a more complex and dense pattern (reticular or circumferential canals) (Reid, 1984; Horner, De Ricqlès & Padian, 1999; Horner, De Ricqlès & Padian, 2000; Horner et al., 2009; Benton et al., 2010; Hübner, 2012; Werning, 2012). Remodeling starts earlier during ontogeny in comparison to the other taxa reported above and widely spaced zones are still present during the sub-adult and adult ontogenetic stages (Reid, 1984; Horner, De Ricqlès & Padian, 1999; Horner, De Ricqlès & Padian, 2000; Horner et al., 2009; Benton et al., 2010; Hübner, 2012; Werning, 2012). Moreover, LAGs count observed in derived iguanodontians are generally lower than that generally found in the more basal taxa reported above, indicating that the somatic maturity was reached earlier (Reid, 1984; Horner, De Ricqlès & Padian, 1999; Horner, De Ricqlès & Padian, 2000; Horner et al., 2009; Benton et al., 2010; Hübner, 2012; Werning, 2012). The disorganized tissue type is still present during the sub-adult and adult ontogenetic stages and the passage to the EFS is abrupt (Reid, 1984; Horner, De Ricqlès & Padian, 1999; Horner, De Ricqlès & Padian, 2000; Horner et al., 2009; Benton et al., 2010; Hübner, 2012; Werning, 2012).

As expected, Ouranosaurus shares similar microstructural patterns with derived iguanodontians. The woven bone and the high vascular density with an alternating reticular and circumferential arrangement present in the bone microstructure suggest fast growth. Remodeling is already present in the sub-adult ontogenetic stage. Fast growth is also supported by widely spaced LAGs with the presence of the same bone structure and type of vascularization within the zones.

Faster growth is phylogenetically coincident with the taxonomical diversification of the derived iguanodontians and their increase in body size (Werning, 2012). It is still unclear whether the higher growth rates are a consequence or a cause of the increase in body size in the clade (Werning, 2012). However, the relatively large body size of Tenontosaurus, coupled with slow growth rates in comparison to those of dryomorphs, suggest that faster growth is a consequence of body size and not the opposite because Ouranosaurus has a length comparable to that of Tenontosaurus, but shows faster growth. As an alternative, Tenontosaurus may represent the maximum size an ornithopod could grow with slow basal growth rates (Werning, 2012).

The function of the back ‘sail’ of Ouranosaurus

Among the Dinosauria, hyperelongation of the neural spines reaches its maximum in the dorsal vertebrae of the theropod Spinosaurus aegyptiacus (see Stromer, 1915; Ibrahim et al., 2014), which also lived in northern Africa during the Cretaceous, although 15–20 million years later than Ouranosaurus. Its neural spines start to elongate from the first dorsal, reaching maximum height at the last dorsal; spine height decreases from the first sacral backwards. Like in O. nigeriensis, neural spines form a sort of a ‘sail’ on the back of the animal. The basal segment of those dorsal neural spines is greatly expanded craniocaudally (Ibrahim et al., 2014). The middle segment is narrow and the apex is expanded with the craniocaudal margins diverging apically, but to a lesser extent than in Ouranosaurus (Ibrahim et al., 2014). The spine height is up to 10 times the centrum height. The apical portion of the spine has sharp cranial and caudal edges; it is marked by thin vertical striae, and is spaced away from adjacent spines (Ibrahim et al., 2014). According to Ibrahim et al. (2014), the ‘sail’ probably had a social function, given the low density of vascularization of the bone pointing against a thermoregulatory function. Recently, new skeletons of the giant ornithomimosaur Deinocheirus mirificus have shown that this weird-looking theropod has also tall neural spines in the middle dorsals (Lee et al., 2014). The neural spines of the proximal dorsals are relatively low, but spine height increases progressively up to the last dorsal, which has a neural spine 8.5 times taller than its centrum height. The base of the neural spines is not craniocaudally expanded as in Spinosaurus. All six sacral neural spines are tall, and, except for the first sacral, the apical parts of the spines are fused into a midline plate of bone with a straight dorsal margin in lateral view (Lee et al., 2014). No hypotheses have been proposed about the function of the ‘sail’ in this dinosaur.

In Ouranosaurus nigeriensis, the elongation of the neural spines is not restricted to the trunk as in Spinosaurus and Deinocheirus, but extends to the proximal caudal region. Occasionally, the base of the neural spine (actually, the prespinal lamina) is slightly expanded in the mid-posterior dorsals like in Spinosaurus, but this expansion occurs only cranially. The apical portions of the neural spines seem to have vertical striations, like in Spinosaurus and Deinocheirus. The external cortex is relatively thinner than in Spinosaurus (Ibrahim et al., 2014) and the spongiosa is thicker. Sharpey’s fibers (which are related to the attachment of muscles and ligaments) occur only in the proximal (basal) part of the neural spine in MSNVE 3714. Based on comparisons with the musculature of crocodiles and birds (Tsuihiji, 2005; Organ, 2006a), this suggests that muscles of the M. transversospinalis group attached just above the base of the neural spines, connecting them at their cranial and caudal edges. The absence of Sharpey’s fibers in the thin sections from the middle and apical portions of the spine, plus the absence of muscle insertion marks on the bone surface may indicate that muscles of the M. transversospinalis group were not attached to the entire surface of the spines. In extant archosaurs, some muscles of the M. transversospinalis group insert on the dorsal margin of the spines; insertion is supposed to occur at the base of the spines in the low-spined synapsid Sphenacodon (Huttenlocker, Rega & Sumida, 2010). In O. nigeriensis, the apparent basal attachment of those muscles may be related to the hyperelongation of the neural spine. However, caution is due because the absence of Sharpey’s fibers and insertion marks could be caused by taphonomic factors as well as preparation and restoration. Ossified tendons have not been found along the vertebral column of the paratype and are reported as scarcely present in the holotype (Taquet, 1976). Ossified tendons along the epiaxial skeleton are an ornithischian general feature. They are usually abundant in iguanodontians and organized in a rhomboidal lattice structure (e.g., Forster, 1990; Organ, 2006a; Organ, 2006b; Norman, 2011; Wang et al., 2011). They are associated with the subunits of M. transversospinalis (Organ, 2006a). Their under-development in Ouranosaurus also supports a peculiar development of the muscles of the M. transversospinalis group in this dinosaur.

In iguanodontian ornithopods, the dorsal neural spines are usually much shorter than those of the theropod Spinosaurus and Deinocheirus. In Iguanodon bernissartensis, the taller neural spines are 2.43 times their centrum height in mid-dorsals (Prieto-Márquez, 2008). Neural spines are proportionally taller in a few other ornithopod taxa, but never as in Ouranosaurus: the spine/centrum height ratio is >4.3 in Morelladon beltrani (see Gasulla et al., 2015), 4.5 in GPIT 1802/1-7 (Iguanodontia indet.; Pereda-Suberbiola et al., 2011), 4.18 in Barbsoldia sicinskii (see Maryanska & Osmólska, 1981) and Hypacrosaurus altispinus (see Prieto-Márquez, 2008).

In Ouranosaurus (MSNVE 3714), the ‘sail’ has a sinusoidal outline, which reaches its maximum height in the mid-proximal dorsals, then decreases towards the sacrum, increasing again in the last sacral and first caudals, and finally decreasing gradually in the rest of the tail (Figs. 1 and 4). In the holotype of Barsboldia sicinskii, which preserves the dorsal, sacral and proximal caudal vertebrae, the outline of the dorsal portion of the ‘sail’ is nearly semicircular and only slightly asymmetrical; the sacral and caudal spines reduce their height gradually moving caudally (Maryanska & Osmólska, 1981). Due to the fragmentary condition of the specimens, the shape of the ‘sail’ is unknown in the other taxa mentioned above. The vertebral column of Hypacrosaurus is still undescribed. The cast of a composite juvenile individual of H. altispinus exhibited at the Royal Tyrrell Museum of Drumheller (Canada) shows a ‘sail’ where the mid-distal dorsal, the sacral and the first caudal spines have the same height (the curvature of the ‘sail’ is actually that of the vertebral column). If that assemblage is reliable, its ‘sail’ is unlike that of Ouranosaurus. At the present state of knowledge, the combination of size and shape of the ‘sail’ of O. nigeriensis is therefore unique.

The only work dealing also with the function of iguanodontian ‘sails’ is that by Bailey (1997), who supported a thermoregulatory role. The tall spines of Ouranosaurus were plausibly a support for a structure like a membrane or a hump (Bailey, 1997). Of course, the definition of such a structure is hampered by the lack of preservation of the soft tissues that were covering the spines. However, the presence of a keratinous covering directly on the bone can be excluded because of the absence of Sharpey’s fibers in the middle and apical portions of the neural spine (Huttenlocker, Rega & Sumida, 2010). Vascularization is not particularly dense in neural spines; this does not support a relationship between elongation and increase of blood input through the bone for thermoregulatory (contra Bailey, 1997) or display purposes. A thermoregulatory role of the ‘sail’ to keep a high constant body temperature as in ectotherm tetrapods would be unnecessary, if the relatively high growth rates observed in Ouranosaurus are correlated with homeothermy. A social (display) role of the structure like that hypothesized for Spinosaurus is possible but, obviously, it is speculative and cannot be tested.

Conclusions

The Venice specimen of Ouranosaurus nigeriensis is mostly made of the paratype GDF 381- MNHN, found in 1970 and collected in 1972 by a French team, although it lacks some of the paratype bones (i.e., the left femur, the right coracoid and a ungual phalanx of manus digit I), which were replaced by plaster copies. Some skeletal elements of the Venice specimen are not present in the field map of the paratype; they could come from a second mapped individual (e.g., some pelvic bones) and possibly from other sources (i.e., the right femur) and were added to the paratype material to complete the mounted skeleton for exhibit purposes.

The Venice specimen shows several differences with the holotype that probably reflect intraspecific variability. Other differences are caused by mistakes in the identification, preparation and reconstruction of the skeletal elements. O. nigeriensis likely had 14 dorsal and one dorsosacral vertebrae, 20 proximal caudal vertebrae, more than the supposed 17 middle caudal vertebrae and a total caudal count much higher than that represented by the preserved caudal vertebrae.

The new information from the Venice specimen and the much higher number of iguanodontian taxa known today with respect to the mid-seventies allows emending of the diagnosis of O. nigeriensis.

The recoding of the character states in the matrices by McDonald et al. (2012b) and Norman (2015) based on the new character information on O. nigeriensis does not change the position of the taxon in the relative phylogenetic hypotheses.

Based on histological analysis, (the first performed on Ouranosaurus), a fast growth rate is assumed for this taxon. The samples show features suggesting a sub-adult ontogenetic stage for the paratype and the other/s individual/s used to assemble MSNVE 3714. Immaturity is suggested also by lack of obliteration of neurocentral sutures and the rough superficial texture of the caudal vertebrae.

Supplemental Information

Supplemental Information 1 Supplementary information on measurements, field maps and phylogenetic analysis

Click here for additional data file.

We thank Mauro Bon, responsible for Research and Scientific Divulgation of the MNSVE, for allowing us access to the specimen under his care; Barbara Favaretto and the staff of the MSNVE for assistance during the study the specimen. Mauro Bon also provided access to the historical information on MSNVE 3714. We acknowledge Philippe Taquet (MNHN) for providing historical information on the paratype and the French expeditions in Gadoufaoua, and Ronan Allain (MNHN) for sending us a copy of the field map of the paratype and the permission to use it. The Fondazione Giancarlo Ligabue (formerly Centro Studi e Ricerche Ligabue) gave us additional information about the Italian expeditions in Gadoufaoua during the 1970s. Karen Poole (Stony Brook University, New York) and Andrew McDonald (St. Louis Science Center, St. Louis) shared information on the codings of Ouranosaurus in their phylogenetic analyses of Ornithopoda. David Norman made available to us the matrix of his phylogenetic analysis, and Andrea Cau provided precious advise on the cladistics. Donald Henderson (TMP) and Albert Prieto-Márquez (The Field Museum, Chicago) are kindly acknowledged for sharing pictures and information about comparative materials. We are indebted to Brooks Britt (Brigham Young University, Provo), Juri Miyamae (Yale University) and Jessica Mitchell (University of Bonn) for the linguistic review of the manuscript. Finally, we thank the reviewers Paul Barrett, Pascal Godefroit, David Norman and the editor Hans-Dieter Sues for their suggestions, which much improved the original manuscript.

Institutional abbreviations

NHMUK Natural History Museum, London (formerly British Museum (Natural History)), UK

CEA Commisariat à l’É nergie Atomique et aux énergies alternatives, France

CNR Comitato Nazionale delle Ricerche, Rome, Italy

CSRL Centro Studi e Ricerche Ligabue, Venice, Italy

GDF field acronym used by French palaeontologists for fossils found in the Gadoufaoua locality

GPIT Geologisches und Palaontologisches Institut der Universität Tübingen, currently the Institut für Geowissenschaften, Tübingen, Germany

IRSNB Institut Royal des Sciences Naturelles de Belgique

MNBH Musée National Boubou-Hama, Niamey, Niger

MNHM Muséum National d’Historie Naturelle, Paris, France

MSNVE Museo di Storia Naturale di Venezia (Natural History Museum of Venice), Venice, Italy

TMP Royal Tyrrell Museum of Palaeontology, Drumheller, Canada

Additional Information and Declarations

Competing Interests

Author Contributions

Data Availability

The authors declare there are no competing interests.

Filippo Bertozzo conceived and designed the experiments, performed the experiments, analyzed the data, wrote the paper, prepared figures and/or tables.

Fabio Marco Dalla Vecchia conceived and designed the experiments, performed the experiments, analyzed the data, contributed reagents/materials/analysis tools, wrote the paper, prepared figures and/or tables, reviewed drafts of the paper.

Matteo Fabbri conceived and designed the experiments, performed the experiments, analyzed the data, contributed reagents/materials/analysis tools, wrote the paper, prepared figures and/or tables.

The following information was supplied regarding data availability:

The raw data is included in the manuscript in the Results section.

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
