# Peer review of "The Venice specimen of Ouranosaurus nigeriensis (Dinosauria, Ornithopoda)"

_PeerJ, doi:10.7717/peerj.3403_

## Round 0.1 · original submission · Major Revisions

· Academic Editor

Major Revisions

This manuscript requires major revision. It contains valuable information but the language of the paper requires a thorough revision by someone who is truly fluent in English. The text is overly wordy, and especially the description needs to be more concise.

·

Basic reporting

The authors have provided a generally clear account, though in some cases their arguments are sometimes difficult to follow due to minor problems with the English (due to slightly awkward phrasing or the use of words whose meaning is slightly inappropriate). I think it would be beneficial for the authors to ask a native English speaker to read through the revised version of their MS to make sure it is clear. This is primarily an issue in the discussion around the history of the Venice Ouranosaurus specimen, which is clearly convoluted, but not always as clear as it could be. The English is sometimes a little colloquial and makes personal comments about the authors of previous papers that could be interpreted as mild insults and these should be removed or paraphrased.

There is some confusion over the use of museum register numbers to refer to specific specimens throughout the study and I think the authors need to make it absolutely clear which numbers currently apply to which specimens to avoid future confusion (see detailed comments below).

The literature cited is sufficient, though the authors have omitted all of the taxonomic references they cite in their Systematic Palaeontology section and there are several other small errors. It would have been nice to see more comparative comments, which if included would require a small expansion in the number of refernences cited.

The article layout is logical and well organised, though I would transfer some of the information on the excavation map to a supplement (see below).

The authors set out their goals clearly and the MS contains all of the information required to support their conclusions.

Experimental design

The authors are to be congratulated on a very useful study that provides much needed information on the rather poorly known Lower Cretaceous iguanodontian dinosaur Ouranosaurus. The paper provides new information on a previously undecribed example of this taxon and the authors provide evidence to support its paratype status, cleaning up a long-standing taxonomic issue and also substantially increasing the amount of basic anatomical data available. The descriptions complement those already available, the figures provide good documentation of the specimen and the osteohistological analysis adds valuable new data on growth in iguanodontians. The research programme was clearly defined and offers novel data. Work has been carried out to a good standard in terms of both the anatomical and histological descriptions, though a little more clarity is needed in the historical account (see below).

I don't have any major ethical concerns, but I am intrigued by the circumstances that led to the specimen appearing in Venice, which seems unusual given it would have originally been the property of the Niger government and I wonder if full transfer of title for this specimen occurred historically and if the museum in Paris was authorised to send this material to another institution outside of Niger, though I am sure this information would be on record in Paris and it would not be encumbent on the authors to prove the Venice museum's ownership of the material.

Methods are clearly explained.

I am slightly surprised that the authors did not take this opportunity to conduct a new phylogenetic analysis or to comment on previous character scores as they have generated new character information from their description. I am also surprised that although they provide a new diagnosis for this taxon they do not discuss the validity/invalidity of previously proposed diagnostic features nor do they provide comparative comments to support the validity of their new diagnosis. I would suggest that the authors should include a new section commenting on the old diagnosis and whether those features were retained or abandoned (and why) and also note how any new features proposed differ from those in other closely related taxa, to support their validity more explicitly.

Validity of the findings

The anatomical and osteohistological descriptions are compentent and seem accurate on the basis of the figures provided. More comparative comments and a few more remarks on what this new specimen adds that was previously unknown (or poorly known) in the holotype would be useful. This MS does add valuable new data.

Conclusions are well supported and based on the information presented in the MS.

The authors are to be congratulated on making it clear where their conclusions are based on solid evidence and where they have had to be more speculative (in terms of the composition of the mounted skeleton, for example).

Additional comments

Detailed comments

1. Although the MS is well written in general and most of the meaning is clear, it would benefit from being checked through by a native English speaker for matters of grammar and style to make the language slightly tighter and to correct a few errors that occasionally obscure or distort meaning. I’ve not gone through the entire MS in detail, but some indications of the kinds of changes required are given in the following examples drawn from the abstract:

Line 16: I suggest “based on” rather than “recorded by”.
Line 16: “a couple” is a very vague term in English that can either mean “two” (formally) or a “small number” (informally). Please say more clearly whether you mean “two” or “several”.
Line 20: Maybe “has been exhibited” rather than “is exposed”?
Line 21: “whether this specimen” rather than “whether it” – makes is clearer you’re referring to the Venice specimen.
Line 24: Maybe “herein the Venice specimen’s” rather than “herein its”.
Line 26: Should be “exception” rather than “exclusion”.
Line 30: “than” rather than “to”.
Line 34: It’s not clear what “sensibly different” means.
Line 35: You could delete the first “The” of this sentence and start it with “Osteohistological”.
Line 36: If you make the abovementioned change, you could also delete “one” from this sentence.
Line 39: Should be “in” rather than “as for”.

2. The section on the history of the specimen is rather confusing – I realise that this is because of a number of mistakes and changes made by earlier authors, but a little additional work to improve the clarity would be helpful to the reader. If I am reading this correctly, this is how I understand the situation:
a) GDF 381 was applied to a skeleton from the second expedition. This skeleton goes on to become the type of Lurdusaurus and is given a new number (GDF 1700) by Russell & Taquet (1999). Is this new number the one that the MNHN still uses for the specimen or do they use the old catalogue number?
b) GDF 381 is erroneously applied to a second specimen (a right coracoid). What is it currently numbered as in the collection?
c) GDF 381 used yet again for a third ornithopod specimen (a skeleton) collected by the fourth expedition. So this specimen, with a number that had been previously applied to two other specimens becomes the paratype as Taquet designated it as such.
So, I think you need to say somewhere, explicitly, that the same field number has been applied in error to three different specimens through the history of these field expeditions (and show how Taquet did this in the 1976 paper with page references to each of the usages – I suspect the information is here already just organised in a way that is difficult to follow through the text). It would be useful to know which specimen (if any) is still currently numbered as GDF 381 (is it Lurdusaurus, the isolated coracoid, or the specimen in Venice? Lurdusaurus seems to have a new number and presumably the Venice specimen has a Venice number, so is it still used for the coracoid?). It is necessary to correct this in the literature so that others can find the right specimens in collections in the future. If necessary you might have to get clarification from the MNHN on what numbers they currently assign to these specimens in their catalogue. On lines 186, 224 and 1333 you mention that the paratype is GDF 381, but this is potentially confusing as it immediately suggests you might be referring to either one of the other two specimens given this number: maybe delete use of the number in each case (if Taquet really does use the same number of several specimens in the same monograph, which isn’t currently clear from your description or whether this history is based both on his monograph and on the MNHN catalogue). On line 188 it would be useful to put the specimen in brackets after each number as well as the year of collection (e.g. GDF 300 – Ouranosaurus holotype; GDF 381, now GDF 1700 – Lurdusaurus holotype). Line 189, I think your descriptions of the Italian expeditions should go into a new paragraph and if there are museum numbers for Italian collected specimens it might be useful to list them (and the material they represent) here. The sentence on lines 219-220 repeats one on 202-207 and could be deleted. In line 241 GDF 381 again used for the paratype even though it could be one of three specimens. Please find a way of referring to the specimen consistently that doesn't involve using this number over and over again if it was given in error following the use of the same number for the specimen that became Lurdusaurus.

3. I’m not sure that the detailed discussion of the difficulties in adding the field map adds much useful information to the paper. Although it is useful to know that a map exists, as it cannot be put together confidently, this is more of an interesting aside than primarily useful in establishing the taphonomy of the site although a briefer version that shows why this confirms the Venice specimen is the paratype would still be useful. The detailed discussion on the difficulties in interpreting the map could be put into a supplementary file. However, it’s not immediately clear to me how the map in itself confirms that the Venice specimen is the paratype. Is this because of the inventory of bones present on the map matches that in the specimen? That the dimentions of the specimen in the field match the Venice specimen? Also, there’s evidence of more than one skeleton here. How confident are you that you can establish which element belongs to which individual? The detail given in the description of the map is much less important than highlighting the evidence that the map provides in identifying the paratype: more discussion of why it ‘proves’ that this skeleton is the paratype is necessary and important as you use the map to justify why some features present in the paratype are better preserved (or in better articulation) than those in the holotype.

4. Specimen numbers. Does the specimen as mounted in Niger have its own specimen number or does that museum simply use the field numbers? Also, rather than use GDF 381 (again!) in line 334 (and line 755 and elsewhere), you should use the Venice number throughout the rest of the paper as that is its current formal museum accession. Using a number that’s been misapplied to various specimens over and over again does not help solve the situation you are trying to clarify! Has the MNHN given new numbers to the other GDF field collections? If so they should be cited here too in line 337.

5. Locality for GDF 301 and 302. Why is it probably the same as that of the holotype (line 344) given that several other localities yielded ornithopod skeletons?

6. Phylogeny. The information given in line 351 onward is interesting, but has no context as you don’t provide any new phylogenetic information. This section could be moved to the general Introduction to the paper (to explain where the taxon sits and current issues in understanding its phylogeny) or, ideally, it should be in a section where you discuss this conflict in relationships in more detail. It’s a shame that you do not use your new information on the specimen to reassess the phylogenetic position, given that the conflict is real and in need of resolution. Does the Venice specimen offer any new character information that could solve this problem? Is there a reason you didn’t reassess the phylogeny in this paper?

7. Lines 376-387. Rather than listing what’s absent it might be a better to list what’s present and let Figure 4 show people what’s missing.

8. Lines 815–816. This information should be in the section on the scapula, not that on the humerus.

9. Lines 867 onwards. Cite the actual paper in which Taquet provides these measurements, rather than just his name.

10. Line 948. “messy” is a rather pointed term that could be interpreted as a personal insult – this sentence should be made more neutral and just make the point that he did not provide a detailed description of the phalanges.

11. I think the plural of manus is actually manüs (Latin, third order declension), not mani.

12. In the section on Differences with the holotype, more detail is needed with this list to make it clear to reader what features the authors regard as biological differences and which are related to taphonomy or damage. The information provided is rather vague at present.

13. It might be useful to provide a table that summarises which elements belong to the paratype, which are reconstructed and which might have been sourced from other specimens.

References

“McDonald” is consistently misspelt throughout the MS.
Wang et al. 2010 is actually 2011 for the final print version (2010 was the online unpaginated version).
The following references are cited in the text, but not listed in the references: Cope 1869; Dollo, 1888; Marsh, 1881; McDonald 2010; Owen, 1842; Reid, 1990; Seeley, 1887; and Sereno, 1997.

·

Basic reporting

This paper is interesting, although it will clearly require a thorough editing by an experienced copy editor (the authors are not native English speakers. Please find attached a version of the ms, in which I have proposed modifications/corrections.

The chapter entitled "The map of the Venice specimen "pro parte" is, in my opinion, particularly problematic. The authors are complainin about the poor quality of the copy of the field map that they received from Ronan Allain, and describe in detail how they tried to reconstruct the original document. Sorry, but it is not interesting at all for the reader. A simpler solution would be to ask for a clearer copy or to consult directly the document in Paris! Please do it before resubmitting the paper.

The paragraph "Phylogenetic relationships of Ouranosaurus nigeriensis" in the Systeamtic Palaeontology should be deleted as the authors do not provide any new information.

I did not understand several sentences or ideas (directly quoted in the attached file.

Experimental design

I really appreciate the holistic aspect of this study. Interestingly, the 'integrity' of a specimen is rarely checked in palaeontological studies. Or, when there is some doubts, the specimen is simply forgotten. Here, the authors were aware of this important problem and tried to solve it using different complementary approaches (checking of all available documents, osteology, osteohistology,...).

I also appreciate the discussion. The authors really tried to make the most of all the data they collected from this specimen. Of course, this is not an exhaustive study, but, thanks to their careful approach of the integrity of the fossil, this specimen can now be included in further studies dealing with different aspects.

But please delete the last paragraph from the conclusion, which is uselessly sermoning (ok for a Master thesis, but not for a scientific paper).

Validity of the findings

Ok for me!

Additional comments

Please checked the attached documents for more comments

·

Basic reporting

1. Poor use of English spelling, grammar and syntax throughout. This should have been copy-edited before submission.
2. Literature refs. Generally OK, but could have been more comprehensive.
3. The article needs to be restructured. At present it is longer than necessary, and rather confused and confusing in content - it needs to be carefully restructured before resubmission.
4. The observations are interesting in a historical context - correcting some of the earlier work of the important and often-used monograph by Taquet (for example). But the interweaving of historical details and anatomical updates leads, ultimately to a confused presentation.
5. The ms needs to be heavily revised. I started to edit some of the basic English for a couple of pages, but then realised that I would have to do a complete re-write to bring this to an acceptable standard of presentation (and that implied quite a lot of restructuring of the account as well). This reads like a thesis in Italian that has been directly transliterated into English - it is too long-winded, tortuous and (in a sense) self-indulgent to be acceptable as it currently stands.

I would strongly recommend a complete re-write of this article. It contains interesting information that deserves to be published, but not in this form.

Experimental design

No comment.
This is self-evident, and referred to above in passing. It needs to be improved considerably.

Validity of the findings

Useful information of relevance to systematists interested in iguanodontian systematics - so, yes, I am interested. But there is still quite a lot of ambiguity - and that should be made as clear as possible because this is an important taxon.

Data is important, but confused and confusing in places.

Conclusions are OK, but better than the actual article (which is the ultimate paradox).

Speculation is confined to the historical narrative for the most part.

Some of the anatomy remains speculative, and this has to be made very clear - because mistakes were made in the original monograph by Taquet.

Additional comments

As above:

Re-write - this needs to be copy-edited by a fluent English speaker
Re-structure - it is confused and confusing as presently structured.
Improve clarity of observations throughout, and ensure that ambiguous anatomical features are make obviously so, so that systematists are not misled, as they have been in the past (through using Taquet as the only source of information about this taxon).

---

## Round 0.2 · Minor Revisions

· Academic Editor

Minor Revisions

The reviewer has carefully edited the manuscript text and offered a list of changes. Please make these changes so that the final version is acceptable for publication.

·

Basic reporting

This version of the MS is much improved on the previous version and the authors have clearly gone to some length to deal with my earlier comments. The presentation is much better, though a large number of minor problems remain with the English usage throughout, though these are all easily fixable and I offer a large number of suggestions for improvements below. References are all appropriate and none obvious are missing, though again there are a couple of minor issues to fix here (see below). The article could still benefit from some careful editing - with a little effort it would be possible to reduce the length of the article by 5-10% without loss by avoiding repetition and, in many places, changing to a terser style. The remit of the paper is well defined and the authors do a thorough and very diligent job in reporting on this specimen.

Experimental design

I've little to add to my previous comments. The phylogenetic analyses look competent and, given that the new information does little to change the systematics of the group, have probably been conducted in sufficient detail, though a little more basic reporting on numbers of trees obtained in each analysis, search protocols used in the TNT analyses, etc. would be useful.

Validity of the findings

All of the authors conclusions seem sound and are supported by the evidence they present. A limited amount of speculation is provided, but this is clearly identified.

Additional comments

Detailed comments, largely grammatical (but with a couple of other comments), follow. I've no additional points to make with respect to the science that weren't covered in my earlier review.

p. 1, line 43. Change “mid-seventies” to “mid-1970s” as this paper may well be cited in other centuries to come and after 2070 this statement would become ambiguous.

p. 2, line 60. Insert ‘of’ between “hypodigm” and “O.”.

p. 2, line 68. McDonald et al. 2012b cited before 2012a. Might need to reorder references to make them appear in the correct order.

p. 2, lines 78–79. This sentence needs slight rewording, maybe: “…can be reliably referred to a single individual or whether it is composed of several individuals.”

p. 2, line 81. Should be “abbreviations”.

p. 2, line 81. The official acronym of the Natural History Museum, London is now NHMUK and has been this for many years (BMNH has been obsolete for nearly two decades, though it is still in widespread and erroneous use). Please update all mentions of BMNH to NHMUK thoughout the text. Moreover, this should read “NHMUK, Natural History Museum, London (formerly British Museum (Natural History)), rather than giving the old name primacy.

p. 2, line 90. The correct spelling is ‘Tyrrell’.

p. 3, line 104. Please insert ‘of’ between “all” and “the”.

p. 3, line 125. If year of publication is the rule determining citation order within the text then Chinsamy (2005) should come between the other two references cited here.

p. 3, line 128. Other then its first use, the three additional uses of the word “the” can be deleted.

p. 3, line 142. As it is a plural “axis” should be ‘axes’.

p. 4, caption for Figure 1. Delete “As” and start the second sentence with ‘For’.

p. 5, line 164. Taquet (1976) and (1998) should be separated by a comma, not a semi-colon.

p. 5, line 166. Delete “the”.

p. 5, line 168. Insert ‘the’ between “of” and “Sahara”.

p. 5, line 168. I think, technically, that the word ‘desert’ should start with a lowercase ‘d’.

p. 5, line 168. Insert ‘in’ between “place” and “January”.

p. 5, line 171. Please substitute ‘labelled’ for “acronymized” (the latter refers to the abbreviation only, not the number also).

p. 6, line 179. Comma missing between authors and year.

p. 6, line 182. Maybe rephrase “remained to” with ‘remained associated with’.

p. 6, line 207. Insert ‘of’ between “all” and “the”.

p. 6, line 208. You could delete “of Paris”.

p. 6, line 209. Substitute “With” with ‘In’.

p. 6, line 224. Substitute “The description …” with ‘A description …’.

p. 6, line 225. Delete “its”.

p. 7, line 230. Rearrange order or the info about the Pers. Comm. so that the date comes last (as in other examples), to avoid this looking like a citation.

p. 7, line 241. Delete “the writing”.

p. 7, line 242. Insert ‘of’ between “all” and “the”.

p. 7, line 245. Insert “with” between “location” and “respect”.

p. 7, line 252. Insert a comma after “column”.

p. 7, line 252. Change “indicate” to ‘indicates’ and “refers” to ‘refer’.

p. 7, line 256. Delete “they”.

p. 8, caption to Figure 2. Line 260 – delete “The” and start sentence with ‘Part’.

p. 8, caption to Figure 2. Line 267. Change “is a handwritten note that refers” to ‘are a handwritten notes that refer”.

p. 8, caption to Figure 2. Line 268. Change “drawn in” to ‘drawn on’.

p. 8, line 275. If using UK spellings, as elsewhere in the MS, this should be ‘PALAEONTOLOGY’.

p. 9, line 295. Change to ‘15 m2 surface’.

p. 9, line 305. “level” should be ‘Level’.

p. 9, line 307. Should be ‘7 km’.

p. 9, line 308. Should be ‘4 km’.

p. 9, lines 308 and 309. You need to include ‘N’ and ‘E’ in your lat/long data to locate it to the correct hemispheres.

p. 9, line 318. “7” should be ‘seven’.
p. 10, line 356. Substitute ‘at’ for “in”.

p. 10, line 359. Delete “the” from after “about”.

p. 10, line 365. Substitute “correlated to” with ‘correlated with’.

p. 10, line 365. Insert ‘and that’ after “quadrupedalism and”.

p. 10, line 366. Delete “as well”.

p. 11, line 376. Suggest rephrase of “has indeed the more elongate skull (length/height ratio >3) within” to ‘does indeed have the most elongate skull (length/height ratio >3) among”.

p. 11, line 377. Change “that” to “than”.

p. 11, line 399. Insert ‘a’ between “such” and “view”.

p. 11, line 400. Delete “the”.

p. 11, line 408. Change “excluded” to ‘excluding’.

p. 11, line 415. Delete “its elongation caused by”, as this is just repetitive.

p. 12, line 417. Might be good to make it clear that you’re referring to the dorsal margin of the lower temporal fenestra in this section.

p. 12, line 453. Change “main axis is” to “main axes are”, as plural.

p. 13, line 479. Insert comma between author name and date.

p. 13, line 480. “Wenhao and Godefroit, 2012” should be ‘Wu and Godefroit, 2012’.

p. 13, line 481. Delete “the”.

p. 13, line 492. Delete “it”.

p. 13, line 494. Not clear if you’re saying that Dryosaurus has 16 or 17 dorsal vertebrae.

p. 13, line 503. Delete “it”.

p. 13, line 505. Substitute ‘on’ for “in”.

p. 13, line 507. Should be ‘five’ not “5”.

p. 14, line 522. Insert a space between “gutter” and the square brackets.

p. 14, line 522. Change “Those” to ‘These’.

p. 14, line 535. Insert comma between author name and date.

p. 15, line 569. Should be ‘styracosternans’.

p. 15, lines 584 and 587. Delete “hind”, as by definition a foot is part of a hind limb.

p. 15, line 585. Substitute “fingers” with ‘toes’.

p. 15, line 589. Delete “as well”.

p. 15, line 596. Delete “the”.

p. 15, line 601. Insert ‘are’ between “as” and “those”.

p. 15, line 611. Insert ‘the’ before “size”.

p. 15, line 615. Insert ‘the’ before “direction”.

p. 15, line 644. Delete “the”.

p. 15, line 652. Insert ‘of’ after “elements”.

p. 15, line 652. Insert “with” before “respect”.

p. 16, caption to Figure 3. Line 658, delete “the”.

p. 16, line 666. Insert “with” before “respect”.

p. 17, line 678. Rephrase “The presacral vertebra 11 has the parapophyses” to ‘Presacral vertebra 11 has parapophyses’.

p. 17, line 680. Change from “The presacral” to “Presacral”.

p. 18, line 691 (and elsewhere). Change “color” to ‘colour’ for consistency and apply throughout.

p. 20, line 721. Change from “From the dorsal vertebra” to ‘From dorsal vertebra’.

p. 20, line 729. Change from “facets of the centra” to ‘facets of centra’.

p. 20, line 731. Change from “straight, inclined” to ‘straight and inclined’.

p. 20, line 739. Change from “and nearly vertical” to ‘and nearly oriented vertically’.

p. 20, line 749. Delete “sensibly”.

p. 20, line 752. Delete “sensibly”.

p. 20, line 755. Delete “the” after “prevent”.

p. 21, line 770. Delete “the”.

p. 22, line 809. Change “craniocaudally” to ‘craniocaudal’.

p. 22, line 823. Change “The dorsal” to “Dorsal”.

p. 23, line 843. Change “surface of the” to ‘surface of’.

p. 23, line 844. Change “and not” to ‘and is not’.

p. 23, line 846. Change “basal one” to ‘basal part’.

p. 23, line 852. Insert “the” before “dorsal”.

p. 23, line 853. Insert “the” before “sacrum”.

p. 24, line 854. Substitute “passage” with ‘transistion’.

p. 25, line 873. Change “missing in” to ‘missing from’.

p. 25, line 886. Insert comma between author name and date.

p. 25, lines 898 and 899. Should be ‘haemapophysis’ for consistency.

p. 25, line 900. “middle and hourglass-shaped” should be ‘middle and are hourglass-shaped”.

p. 25, line 901. “of the caudals” should be ‘of caudals’.

p. 25, lines 904–905. Rephrase from “seems to appear in correspondence of the shift of the haemapophyseal facets on the” to ‘seems to correspond with the shift of the haemapophyseal facets on to the’.

p. 25, line 911. Change “scarcely projecting” to ‘scarcely project’.

p. 26, line 921. Change “with cranial-facing” to ‘with a cranial-facing’.

p. 26, lines 922–923. Rephrase from “portion of some neural spine was broken” to ‘portions of some neural spines were broken’.

p. 26, line 926. Insert comma between author name and date.

p. 26, line 938. Delete “both preserved distally”.

p. 26, line 941. Change “As for” to ‘As with’.

p. 26, line 942. Change “are less” to ‘is less’.

p. 26, lines 946–948. Insert commas between author names and dates.

p. 26, line 957. Change “While rarity” to ‘While the rarity’.

p. 26, line 961. Organ 2006b cited, but there has been no prior reference to Organ 2006a.

p. 27, caption to Figure 11. “Evidences” should be “Evidence”.

p. 29, line 1026. Insert “with” before “respect”.

p. 30, line 1047. Change “with the metacarpal” to ‘with metacarpal’.

p. 30, line 1056. Change “to the metacarpal” to ‘to metacarpal’.

p. 31, line 1058. “wrist” should be ‘wrists’.

p. 31, line 1064. “relative” should be ‘relevant’.

p. 31, line 1078. Change “convex for” to ‘convex in’.

p. 31, line 1084. Substitute “leans on it” with ‘abuts it’.

p. 31, line 1090. Change “extremities has” to ‘extremities and has’.

p. 31, line 1092. Change “who probably refers to the” to ‘who is probably referring to’.

p. 31, line 1106. Should be ‘suggests that’.

p. 34, line 1189. Insert “with” before “respect”.

p. 34, lines 1190 and 1191. The words ‘brevis’ and ‘brevis fossa’ do not need to be italicized.

p. 36, line 1233. Change “bases” to ‘based’.

p. 36, lines 1256–1257. Insert commas between author names and dates.

p. 38, line 1274. Change “presents” to ‘contains’.

p. 38, line 1276. Change “Astragalus and calcaneum are glued to tibia and fibula” to ‘The astragalus and calcaneum are glued to the tibia and fibula’.

p. 38, line 1300. Change “all the” to ‘all of the’.

p. 38, line 1308. Change “long as wide” to ‘long as it is wide’.

p. 38, line 1310. Change “much” to ‘and much’.

p. 40, lines 1343–1344. Rephrase “We performed the cladistic analysis of O. nigeriensis using the new information from the Venice specimen to reassess its phylogenetic position, as requested by one of the referees” to ‘Using this new information from the Venice specimen, we performed a cladistics analysis to reassess the phylogenetic position of Ouranosaurus nigrensis’.

p. 40, line 1345. Delete “potentially’.

p. 40, lines 1347–1350. Rephrase “The character list (journal.pone.0036745.s003), taxa and scores (journal.pone.0036745.s001) from McDonald (2012; "second run" on p. 2) were taken to create a Nexus file matrix. Matrix and taxa were emended according to McDonald et al. (2012b), adding the codings for Proa valdearinnoensis” to “We used the data matrix of
McDonald (2012; "second run" on p. 2) with the addition of scores from McDonald et al. (2012b) for Proa valdearinnoensis’.

p. 40, lines 1343–1350. Somewhere in this section you need to mention the number of taxa and characters that you have in your new data matrix.

p. 40, line 1366. You should give a reference for TNT (Goloboff et al. 2008) and should also mention the search parameters you used in this analysis and the statistics for the tree(s) you obtained, as these sometimes differ between TNT and PAUP.

p. 40, line 1380. Change “The revision of the whole phylogenetic” to ‘Revision of the phylogenetic’.

p. 42, lines 1397–1399 can be deleted as these just repeat information.

p. 42, line 1403. Delete “was”.

p. 44, Figure 21 did not convert into the PDF.

p. 44, line 1458. Delete “as’.

p. 46, line 1487. Suggest changing “thinner and thinner” for ‘progressively thinner’.

p. 48, line 1530, Should be ‘cross-section’, apply throughout.

p. 49, line 1544. Should be ‘craniolateral’. I also don’t understand what “intersect” means – do you mean ‘cross-section’?

p. 50, line 1564. Substitute “integrate” with ‘complete’.

p. 50, line 1601. Substitute “couples” with ‘pairs’.

p. 50, line 1602. Substitute “couple” with ‘pair’.

p. 51, line 1623. Change “as it” to ‘as if it’.

p. 51, line 1643. Change “Different length” to ‘The different length’.

p. 51, line 1644. Change “who is” to ‘that is’.

p. 51, line 1646. Change “occurs” to ‘is preserved in’.

p. 51, lines 1646–1647. Change “The humerus, the tibia and the dorsal rib” to ‘The
humerus, tibia and dorsal rib’.

p. 52, lines 1659–1660. Change “which the humerus, the tibia, the neural spine and the dorsal rib come from.” to ‘which the humerus, tibia, neural spine and dorsal rib come from.’.

p. 52, line 1665. Insert ‘the’ before “astragalus”.

p. 52, line 1672. Delete “instead”.

p. 52, line 1689. Change “all the” to ‘all of the’.

p. 52, line 1701. Change “A possible evidence” to ‘Possible evidence’.

p. 52, line 1702. Change “correspondant” to ‘corresponding’.

p. 53, line 1708. Should be ‘by undertaking’.

p. 53, line 1708. Change “it results” to ‘it is shown’.

p. 53, line 1711. Should be ‘7 m’.

p. 52, line 1716. Should be ‘the total length’.

p. 52, line 1717. Sentence should start ‘The humerus …’.

p. 53, line 1757. Should be “from the” not “with the”.

p. 53, line 1757 onwards. Almost this entire section repeats information you’ve given elsewhere or commented on previously. Is this really necessary? I would condense it or delete it.

p. 53, line 1759. Should be ‘as also reported’.

p. 53, line 1761. This sentence should start ‘These include’, not “Such are”.

p. 56, line 1860. Should be ’10 times’.

Reid, 1990 is cited in the text, but isn’t in the reference list.
Sereno, 1997 is in the references, but not cited in the text.
Taquet, 1970 is in the references, but not cited in the text.
You et al., 2005 is cited in the text, but isn’t in the reference list.
You et al. 2009 is in the references, but not cited in the text.

Some of the references do not run in correct date or alphabetical order (e.g. Taquet, 1975 is listed after 1976, 1998).

---

## Round 0.3 · accepted · Accept

· Academic Editor

Accept

The revised version has properly addressed all requested changes.